# PTEN restrains SHH medulloblastoma growth through cell autonomous and nonautonomous mechanisms

Zhimin Lao[1], Salsabiel El Nagar[1†‡], Yinwen Liang[1†§], Daniel Stephen[1], Alexandra L Joyner[1,2*]

[1]Developmental Biology Program, Sloan Kettering Institute, Memorial Sloan Kettering Cancer Center, New York, United States; [2]Biochemistry, Cell and Molecular Biology Program, Weill Cornell Graduate School of Medical Sciences, New York, United States

**\*For correspondence:**
joynera@mskcc.org

[†]These authors contributed equally to this work

**Present address:** [‡]Mangrove Neuro, Long Island City, NY, United States; [§]Capital Medical University and Chinese Institutes for Medical Research, Beijing, China

## eLife Assessment

This **valuable** study provides insights into the role of Pten mutations in SHH-medulloblastoma, by using mouse models to resolve the effects of heterozygous vs homozygous mutations on proliferation and cell death throughout tumorigenesis. The experiments presented are **convincing**, with rigorous quantifications and orthogonal experimentation provided throughout, and the models employing sporadic oncogene induction, rather than EGL-wide genetic modifications, represent an advancement in experimental design. However, additional experimentation focused on a greater characterization of macrophage phenotypes (e.g., microglia vs circulating monocytes) would enhance this study. The work will be of interest to medical biologists studying general cancer mechanisms, as the function of Pten may be similar across tumor types.

**Abstract** A third of patients with the pediatric cerebellar tumor Medulloblastoma (MB) have mutations that activate Sonic hedgehog (SHH) signaling (SHH-MB subgroup). The contribution of secondary mutations to tumor severity, however, is not clear. *PTEN* mutations are enriched in the SHH-1 subtype that has the lowest survival rate. Widespread heterozygous loss of *Pten* in two SHH-MB mouse models increases penetrance and accelerates onset of differentiated tumors. We delineated cellular and transcriptional changes that accelerate tumor growth and cause differentiation using a sporadic SHH-MB mouse model expressing oncogenic SmoM2 in rare cerebellar granule cell precursors (GCPs) and scRNA-seq analysis. Homozygous but not heterozygous sporadic loss of *Pten* resulted in rapid acceleration of tumor growth and end-stage disease by 40 days, compared to ~25% survival in control SmoM2 mice at 100 days. Heterozygous *PTEN* mutations, therefore, should negatively impact disease outcome primarily with germline mutations. Loss of *Pten* in normal or SmoM2-expressing GCPs increased proliferation and enhanced progenitor state initially, but by 12 days *Pten* mutant SmoM2 tumors were highly differentiated due to increased survival of non-proliferating GCPs. Furthermore, macrophage infiltration and cytotoxicity appeared reduced in differentiated regions of tumors lacking *Pten*, indicating cell nonautonomous changes could also contribute to accelerated tumor growth.

## Introduction

Medulloblastoma (MB) is a brain tumor seen mainly in pediatric patients and which has four molecular subgroups, WNT, SHH, Group 3, and Group 4 (*Taylor et al., 2012*). The Sonic hedgehog subgroup of

MB (SHH-MB) constitutes approximately 30% of MB and is characterized by mutations that constitutively activate SHH signaling. Most SHH-MBs have one of four types of mutations that activate SHH signaling: activating mutations in *SMO*, loss-of-function mutations in the pathway inhibitors *PTCH1* and *SUFU,* or loss of *TP53* in association with amplification of the pathway effectors *GLI2* or *MYCN*. SHH-MB is further divided into four subtypes based on molecular, biological, and clinical parameters (*Cavalli et al., 2017*; *Kool et al., 2014*; *Robinson et al., 2018*; *Schwalbe et al., 2017*; *Garcia-Lopez et al., 2021*). The work on SHH-MB subtypes has highlighted that particular secondary mutations likely contribute to tumor severity. For example, heterozygous *PTEN* and *KMT2D* loss-of-function mutations are enriched in the SHH-1 or beta subtype, which has the lowest overall survival rate and highest frequency of metastasis and occurs primarily in infants (<4 years). Our recent work on the chromatin modifier *KMT2D* using two SHH-MB mouse models showed that when one or two copies of *Kmt2d* are deleted, not only is there a nearly twofold increase in tumor penetrance with a Classic tumor cytoarchitecture (highly proliferative), but also strikingly a near 100% penetrance of spinal cord metastasis and hindbrain invasion (*Sanghrajka et al., 2023*). The transcriptional/chromatin landscape of mouse *Kmt2d* mutant SHH-MB primary and metastatic tumors was found to be shifted towards decreased expression of neural differentiation genes and an increase in oncogenes seen in more aggressive tumors compared to tumors with intact *Kmt2d* (*Sanghrajka et al., 2023*). Thus, mouse models support the clinical data that loss of *KMT2D* contributes to more aggressive disease.

The gene encoding the dual specificity phosphatase *PTEN* (Phosphatase and TENsin homolog deleted on chromosome 10) is one of the most frequently mutated tumor suppressor genes in human cancers (*Lee et al., 2018*) and is mutated in ~5% of SHH-MB (*Kool et al., 2014*). PTEN regulates an array of cellular functions, including cell survival, proliferation, and metabolism through attenuating phosphoinositide-3 kinase (PI3K-AKT) signaling and genome stability through nuclear functions. Recent work showed that PTEN also can be secreted and act in a cell nonautonomous manner (*Hopkins et al., 2013*; *Putz et al., 2012*; *Zhang et al., 2024*). PTEN is secreted by tumor cells via binding to TMED10 and then binds to PLXDC2 on tumor-associated macrophages (TAMs) and induces inflammatory status through triggering JAK2-STAT signaling, thus contributing to the tumor suppressor function of PTEN. Mouse models of SHH-MB indicate loss of *Pten* increases tumor incidence, but the underlying cell autonomous and nonautonomous reasons and impact on metastasis are not as clear (*Castellino et al., 2010*; *Metcalfe et al., 2013*).

SHH-MB arises in *Atoh1*-expressing granule cell precursors (GCPs) on the surface of the developing cerebellum. These precursor cells are dependent on SHH signaling for their sustained proliferation and major expansion during the first two weeks after birth in mice (*Corrales et al., 2006*; *Lewis et al., 2004*; *Joyner and Bayin, 2022*). In a mouse model of SHH-MB in which a *Neurod2* promoter drives expression of an oncogenic form of SMO (SmoA1) in the GCP-lineage, heterozygous loss of *Pten* in all cells (null mutation) or conditional knock-out in Nestin-expressing cells, which includes all cerebellar cells (*Tronche et al., 1999*), leads to increased penetrance and earlier onset of tumor formation with no increase in metastasis (*Castellino et al., 2010*). Whereas most mouse models of SHH-MB have a classic (highly proliferative) histology, the *Pten* heterozygous mutant tumors have a nodular character with extensive regions of neural differentiation (NeuN + cells) and an apparent decrease in cell death and increase in angiogenesis. In a mouse SHH-MB conditional model in which *Ptch1* is homozygously deleted at embryonic day 14 (E14) in all GCPs (*Atoh1-Cre*), additional homozygous loss of *Pten* leads to rapid tumor formation, and heterozygous loss leads to a slightly more aggressive phenotype than *Ptch1* loss alone (*Metcalfe et al., 2013*). Similar to *Neurod2*-SmoA1 tumors, in *Ptch1* mutant tumors, *Pten* loss causes highly differentiated tumors with decreased cell death. In a mutagenic screen for increased SHH-MB penetrance and metastasis using a Sleeping Beauty transposon approach in mice with heterozygous mutations in *Ptch1* or *Trp53*, insertions found in tumors with spinal cord metastasis were associated with mutations in *Pten*, *Akt2*, *Pik3r1,* and *Igf2*, implicating increased PI3K signaling in metastasis (*Wu et al., 2012*). Consistent with this, in a mouse model where *Shh* is expressed in GCPs, additional expression of an activated form of AKT increases tumor incidence and spinal cord metastasis (*Wu et al., 2012*; *Mumert et al., 2012*). Data from human tumors support that a reduction in *PTEN* protein enhances tumor progression, since SHH-MB human patients with reduced or no PTEN protein have a worse overall survival (*Castellino et al., 2010*). However, no correlation was seen between PTEN protein level and histology or metastasis. Thus, several questions remain as to whether *Pten* loss increases metastasis, what accounts for the faster growth and a more differentiated cellular

phenotype in mouse *Pten* mutant SHH-MBs, and why only in some models *Pten* heterozygous loss is sufficient to greatly enhance tumor progression.

We addressed these questions by comparing tumor progression and spinal cord metastasis in a sporadic mouse model of SHH-MB in which rare GCPs are induced to express an oncogenic form of SMO, SmoM2, at postnatal day 0 (P0) and have deletion of one or both copies of *Pten*. We found that only homozygous loss of *Pten* combined with SmoM2 expression enhances tumor progression in this sporadic model, and metastasis is not increased. Furthermore, the differentiated phenotype of the tumors is primarily due to a specific decrease in cell death of non-proliferative cells in the tumors, whereas cell death is as high in regions of high proliferation as in SmoM2-expressing tumors with normal *Pten* or heterozygous loss of *Pten*. Loss of *Pten* in normal GCPs, or those expressing SmoM2, leads to a mild initial increase in the proliferation index, indicating a cell autonomous function of PTEN in reducing proliferation. Single-cell RNA-sequencing (scRNA-seq) analysis revealed that proliferating tumor cells that lack *Pten* not only have the expected increase in mTOR signaling, but also have altered expression of genes involved in neural differentiation and downregulation of inflammatory and interferon alpha response genes. Homozygous loss of *Pten* in GCPs also leads to a major decrease in infiltration of macrophages into tumors and an altered transcriptome indicating reduced cytotoxicity, revealing a possible cell nonautonomous function of PTEN that normally reduces tumor progression.

## Results

### *PTEN* mutations co-occur with mutations that activate SHH-signaling in human SHH-MB

Using the cBioPortal (*Gao et al., 2013*; *Cerami et al., 2012*) and a dataset of 127 patient samples considered to be SHH-MB (*Kool et al., 2014*), we determined whether patients with mutations that activate SHH-signaling have *PTEN* mutations. 4.3% of the 127 samples had a mutation in *PTEN* and furthermore 4/5 of the mutations co-occurred with mutations that activate SHH-signaling (*PTCH1*, *SUFU,* or *SMO*; *Figure 1A*). Thus, although the *PTEN* mutations were detected in a low percentage of patient tumors, *PTEN* appears to be predominantly mutated in SHH-MB along with mutations that activate SHH-signaling.

### Homozygous loss of *Pten* in sporadic SHH-MB accelerates growth and increases penetrance to 100%

We next used our mosaic mutant technique (MASTR; *Lao et al., 2012*) to test the impact of lowering *Pten* in a sporadic mouse model of SHH-MB. The approach involves inducing expression of an oncogenic form of the SHH receptor SMO (SmoM2) and eGFP in ~20,000 GCPs (~0.1%) at P0 (*Figure 1B*; *Lao et al., 2012*; *Tan et al., 2018*; *Mao et al., 2006*). The model allows penetrance and survival time to be determined since less than 75% of mutant mice die by 100 days. Using the model, we asked whether loss of one or both copies of *Pten* (*Trotman et al., 2003*) in rare GCPs-expressing SmoM2 changes the survival curves of mice by comparing *Atoh1-FlpoER/+; R26$^{MASTR/LSL-SmoM2}$* mice (referred to as SmoM2) to *Atoh1-FlpoER/+; R26$^{MASTR/LSL-SmoM2}$; Pten$^{fl/fl}$* (referred to as SmoM2-*Pten$^{fl/fl}$*) and *Atoh1-FlpoER/+; R26$^{MASTR/LSL-SmoM2}$; Pten$^{fl/+}$* (referred to as SmoM2-*Pten$^{fl/+}$*) mice administered tamoxifen at P0 (*Figure 1B*). Strikingly, all SmoM2-*Pten$^{fl/fl}$* mice succumbed to disease between P20 and P40, whereas SmoM2 and SmoM2-*Pten$^{fl/+}$* mice began dying at ~P40 and by P100, ~25% of mice of both genotypes survived (*Figure 1C*). Thus, loss of both *Pten* alleles is required to accelerate tumor progression in a mosaic SmoM2 model of SHH-MB, unlike a *Neurod2*-SmoA1 model with germline or broad neural deletion of *Pten* in the cerebellum where heterozygous deletion is sufficient (*Castellino et al., 2010*).

### Homozygous and heterozygous loss of *Pten* does not increase metastasis to the spinal cord in SmoM2 SHH-MB

Given the different reported findings as to whether *Pten* loss increases metastasis in mouse models of SHH-MB, we asked whether in our sporadic SmoM2 mice loss of *Pten* alters the frequency of spinal cord metastasis when mice show signs of brain tumor disease. Coronal spinal cord sections of each mouse (n=55–138 sections per mouse) were analyzed by cytology for the presence of spinal cord metastasis and tumors were confirmed with immunohistochemical (IHC) staining for GFP, Ki67, and PTEN (*Figure 1—figure supplement 1A–C*). Quantification of the percentage of mice that had at

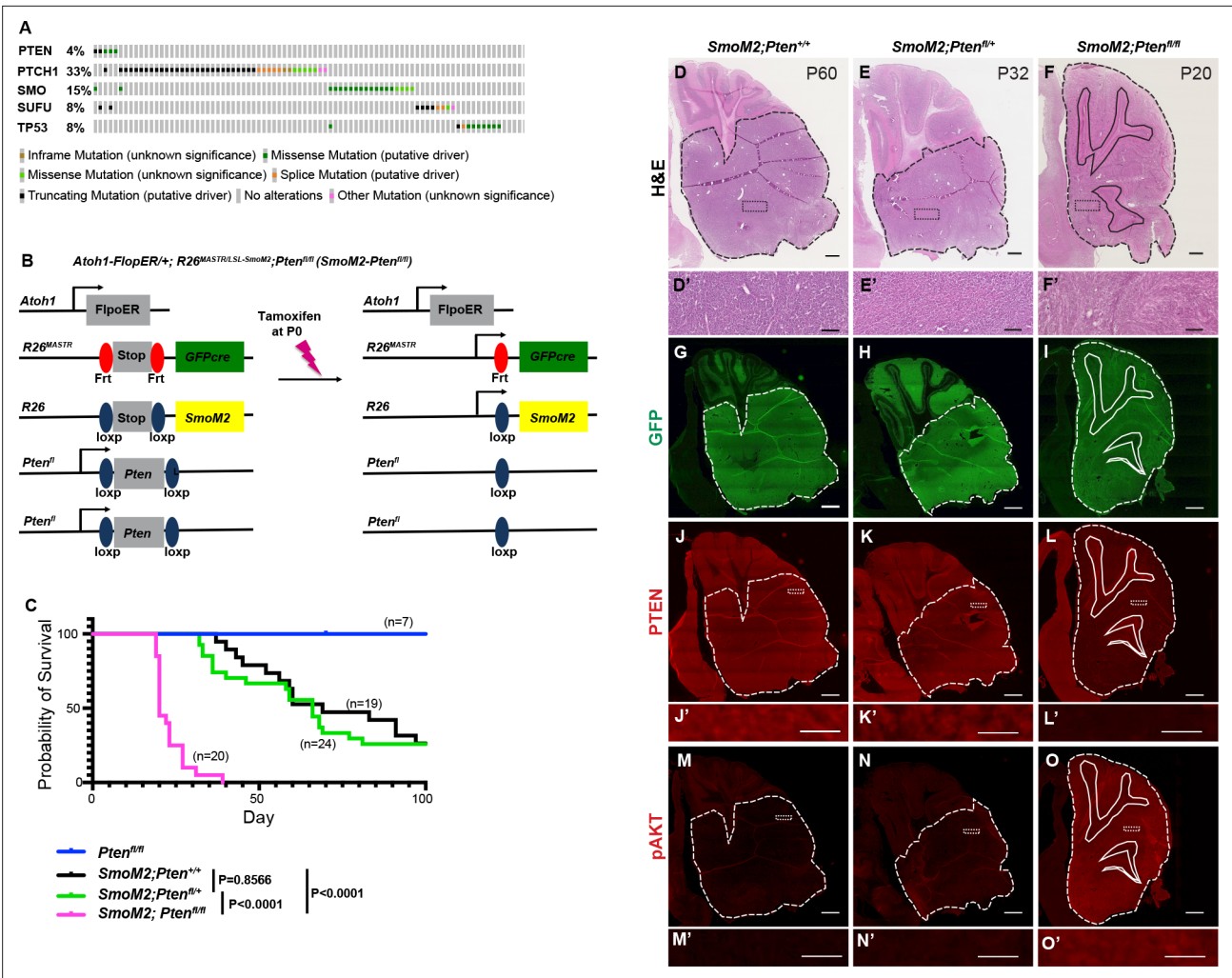

**Figure 1.** Homozygous but not heterozygous loss of *Pten* in a sporadic mouse model of SmoM2 SHH-MB increases penetrance and reduces time of disease onset. (**A**) An Oncoprint of 27 patient samples considered to be SHH-MB (*Kool et al., 2014*) using the cBioPortal (*Gao et al., 2013*; *Cerami et al., 2012*) showing the frequency of mutations in *PTEN* and genes that activate the SHH pathway showing co-occurrences. (**B**) Schematic showing Mosaic mutant Analysis with Spatial and Temporal Regulation (MASTR) technique used to model SHH-MB by expressing SmoM2 in scattered GCPs starting at P0 and deleting one or two copies of *Pten* (*Atoh1-FlpoER/+*; *R26^{MASTR/LSL-SmoM2}*; *Pten^{fl/fl}* or SmoM2-*Pten^{fl/fl}* mice). (**C**) Kaplan-Meier curve and statistics (below) for survival of SmoM2 mice compared to SmoM2-*Pten^{fl/fl}* and SmoM2-*Pten^{fl/+}* mice. Statistics were calculated using the log-rank test. (**D–F**) Representative images of hematoxylin and eosin (H&E) stained sagittal sections of end stage tumors from SmoM2, SmoM2-*Pten^{fl/+}*, and SmoM2-*Pten^{fl/fl}* mice with higher magnification images shown below (n>8 per genotype). Scale bars indicate 1 mm (**D–F**) and 100 µm (**D'-F'**). (**G–I**) GFP-stained sagittal sections of end-stage tumors from the indicated mutants (n=5–8 per genotype G-O). Scale bar indicates 1 mm. (**J–L**) Sagittal sections of end-stage tumors from the indicated mutants stained for PTEN (**J–L**) or pAKT (**M–O**) with higher magnifications below. Scale bars indicate 1 mm (**J–O**) and 20 µm (**J'-O'**). (**D–O**) Dashed lines outline the tumors, solid lines outline IGL, dotted rectangles indicate location of high-magnification images.

The online version of this article includes the following figure supplement(s) for figure 1:

**Figure supplement 1.** Loss of *Pten* does not increase metastasis to the spinal cord in SmoM2 SHH-MB.

least 2 sections with more than a few tumor cells showed a similar tumor incidence between the three genotypes (SmoM2: 64%, n=11; SmoM2-*Pten^{fl/+}*: 69%, n=13; SmoM2-*Pten^{fl/fl}*: 58%, n=12). Furthermore, quantification of the percentage of spinal cord sections per mouse with tumor cells showed no difference between the *Pten* mutant tumors and *SmoM2* (*Figure 1—figure supplement 1D*). Likewise, measuring the area of the tumors did not detect a difference in metastatic disease when *Pten* was lowered or deleted (*Figure 1—figure supplement 1E and F*). Thus, in mouse sporadic SmoM2 SHH-MB, heterozygous or homozygous loss of *Pten* does not appear to increase spinal cord metastasis.

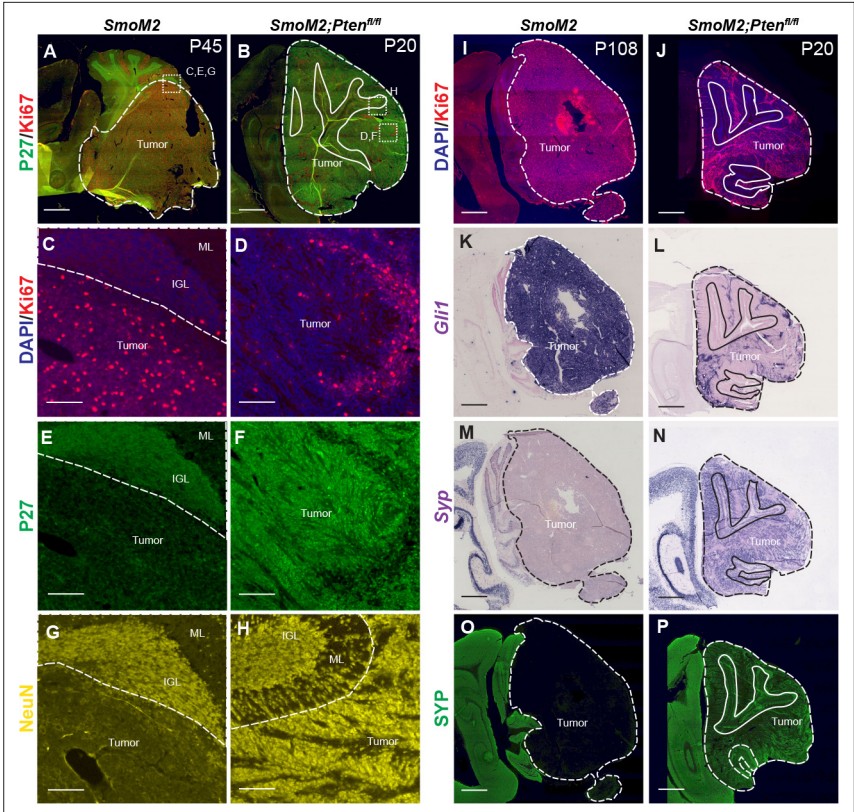

**Figure 2.** *Pten* loss in SmoM2 sporadic SHH-MB induces highly differentiated tumors. (**A, B**) Sagittal sections of the cerebellum showing end stage tumors from SmoM2 and SmoM2-*Pten*^fl/fl mice stained for P27 (post-mitotic cells) and Ki67 (proliferating cells; SmoM2, n=5; SmoM2-*Pten*^fl/fl n=7). Dashed lines outline each tumor, solid lines outline IGL, dotted squares indicate location of high-magnification images in (**C–H**). (**C–H**) High power images of sections stained for Ki67 and DAPI (**C–D**), P27 (**E–F**) or NeuN (**G–H**) (SmoM2, n=5; SmoM2-*Pten*^fl/fl n=7). Dashed line separates the internal granule layer (IGL) or molecular layer (ML) from the tumor. (**I,J**) Sagittal sections of the cerebellum showing lateral tumors from SmoM2 and SmoM2-*Pten*^fl/fl mice stained for Ki67 and DAPI (SmoM2, n=7; SmoM2-*Pten*^fl/fl n=8). (**K–N**) RNA in situ hybridization staining of sections for *Gli1* (**K–L**) and *Syp* mRNA (**M–N**) (n=3 per genotype). (**O–P**) Immunohistochemical staining of sections for SYP protein (n=6 per genotype). (**I–P**) Dashed lines outline the tumors, solid lines outline IGL. (**A–B, I–P**) Scale bars indicate 1 mm. (**C–H**) Scale bars indicate 100 µm.

The online version of this article includes the following figure supplement(s) for figure 2:

**Figure supplement 1.** Spontaneous loss of the second allele of *Pten* in rare cells of some SmoM2-*Pten*^fl/+ tumors.

## *Pten* loss in a SmoM2 sporadic mouse model of SHH-MB induces highly differentiated tumors

Hematoxylin and eosin (H&E) staining of primary tumor sections indicated that homozygous loss of *Pten* results in tumors with decreased cellularity (*Figure 1D–F*). IHC staining of tumor sections at end stage confirmed that most tumor cells expressed GFP (*Figure 1G-I*) and only SmoM2-*Pten*^fl/fl tumors lacked PTEN (*Figure 1J-L*) and had strong Phospho-AKT (pAKT; *Figure 1M-O*) and pS6 staining (*Figure 2—figure supplement 1A–D*), thus confirming absence of PTEN and activation of mTOR (mammalian target of rapamycin) signaling only in SmoM2-*Pten*^fl/fl tumors. Staining with markers of cell proliferation (Ki67) and differentiation revealed that whereas SmoM2 tumors consisted of mostly Ki67 + cells, SmoM2-*Pten*^fl/fl tumors were highly differentiated, expressing the post-mitotic marker P27 and neural differentiation marker NeuN, with only small areas of proliferative cells (*Figure 2A–H*). The SmoM2 tumors had broad expression of the SHH target gene *Gli1*, whereas the SmoM2-*Pten*^fl/fl tumors expressed *Gli1* mRNA only in Ki67 + regions of the tumors (*Figure 2I–L*), which was complementary to the neural differentiation marker Synaptophysin (SYP/*Syp*) that was absent from SmoM2 tumors (*Figure 2K–P*). Thus, despite SmoM2 being expressed in the differentiated cells of the

SmoM2-Pten^fl/fl tumors, SHH signaling is downregulated specifically in differentiated cells. Interestingly, in 3/9 tumors from SmoM2-Pten^fl/+ mice some sections had patches of cells devoid of PTEN and positive for pAKT, indicating spontaneous loss of the second allele of *Pten* in rare cells that expand as a coherent group (*Figure 2—figure supplement 1E–H*). The regions without PTEN in SmoM2-Pten^fl/+ mice had a similar differentiated phenotype (SYP +and NeuN+) to SmoM2-Pten^fl/fl tumors (*Figure 2—figure supplement 1I–K*).

### *Pten* loss protects differentiated tumor cells from cell death

Mouse SHH-MB models that have a highly differentiated cytology are rare and have primarily been reported in two *Pten* models (*Castellino et al., 2010*; *Metcalfe et al., 2013*). In a *Ptch1* mutant model of SHH-MB, it was shown that proliferating tumor cells give rise to NeuN +postmitotic tumor cells that then die at a high rate (*Vanner et al., 2014*). We therefore asked whether *Pten* loss (increased mTOR signaling) rescues the post-mitotic tumor cells from death by staining for SYP or NeuN and TUNEL (*Figure 3A–H*). SmoM2-Pten^fl/fl tumors had significantly less TUNEL particles in regions of high SYP labeling (SYP Hi) compared to SYP low regions (*Figure 3C–I*, *Figure 3—figure supplement 1*). Moreover, in the regions of low SYP, the density of TUNEL staining was similar to in SmoM2 tumors where SYP is low throughout the tumor (*Figure 3J*). As expected, in PTEN-negative patches of SmoM2-Pten^fl/+ tumors, TUNEL staining was lower in these SYP Hi regions of differentiated cells (NeuN+) than in SYP low proliferative regions (Ki67+; *Figure 3J and K*, *Figure 2—figure supplement 1I-L*).

### Loss of *Pten* leads to rapid cell autonomous changes in SmoM2 GCP behaviors

Given how rapidly the tumors grow in the SmoM2-Pten^fl/fl mice and that lethal disease occurs between P20-40, we asked how early a differential growth phenotype can be seen in the three genotypes. In our previous study of the effect of loss of *Kmt2d* in SHH-MB models, no difference in tumor size was detected at P12 but was by P21 (*Sanghrajka et al., 2023*). We therefore asked whether SmoM2-Pten^fl/fl tumors have a growth advantage over *SmoM2* tumors at P12. Staining of sagittal sections of SmoM2 tumors of the three *Pten* genotypes with EdU (1 hr pulse) and GFP to mark the proliferating GCPs in the external granule cell layer (EGL) showed a massive expansion of the EGL in SmoM2-Pten^fl/fl mice compared to SmoM2 mice (*Figure 4A–C*). Quantification of cerebellar area in midline sections revealed a significant increase in SmoM2-Pten^fl/fl mice compared to SmoM2 mice and SmoM2-Pten^fl/+ mice (*Figure 4D*). The EGL area alone showed a fourfold average area increase in SmoM2-Pten^fl/fl mice compared to SmoM2 mice, and SmoM2-Pten^fl/+ mice had a slight increase (<2-fold) compared to SmoM2 mice (*Figure 4E*). As in end-stage tumors, the tumors in SmoM2-Pten^fl/fl mice had a significant decrease in dying cells compared to the other two genotypes at P12 (*Figure 4F-L*, *Figure 4—figure supplement 1*). In large lesions with extensive differentiation (NeuN +Ki67 cells), the density of TUNEL particles was decreased in NeuN +regions compared to proliferative regions. As expected, pAKT staining was upregulated in the EGL of SmoM2-Pten^fl/fl mice with no detectable staining in the other two genotypes (*Figure 4M–O*).

One prediction of our findings is that once SmoM2 GCPs lacking *Pten* differentiate, their survival should be enhanced. Therefore, if proliferating GCPs are labeled and then fate mapped for 2 days, a higher percentage of SmoM2-Pten^fl/fl labeled cells should be differentiated (post-mitotic) compared to in SmoM2 mice. To test this idea, we administered one injection of BrdU into P10 mice and then sacrificed them at P12 and quantified the percentage of BrdU + cells in the EGL that were Ki67- (*Figure 4P*). As predicted, a higher percentage of BrdU + cells in SmoM2-Pten^fl/fl mice were Ki67- at P12 compared to SmoM2 (*Figure 4Q*). This result indicates that by P10 after loss of *Pten* at P0, either differentiation is promoted in SmoM2 GCPs and/or that the differentiated progeny of GCPs lacking PTEN are spared from cell death. The proliferation rate of GCPs in SmoM2 and SmoM2-Pten^fl/fl mice was similar at P12 (*Figure 4R*), as determined by injecting EdU 1 hr before sacrificing the mice (%EdU +and Ki67 + cells/Ki67 + cells), indicating the latter possibility.

Given that at P12 the cellular phenotype of GCPs in SmoM2-Pten^fl/fl mice was similar to in mice at end stage, we analyzed GCPs at P8, 8 days after the mice were treated with tamoxifen. Similar to at P12, the area of the proliferating EGL of the cerebellum was significantly greater in both the SmoM2-Pten^fl/fl and the SmoM2-Pten^fl/+ mice compared to SmoM2, although the area of the cerebellum was similar across genotypes, likely due to the relatively small increase in EGL area (~2-fold)

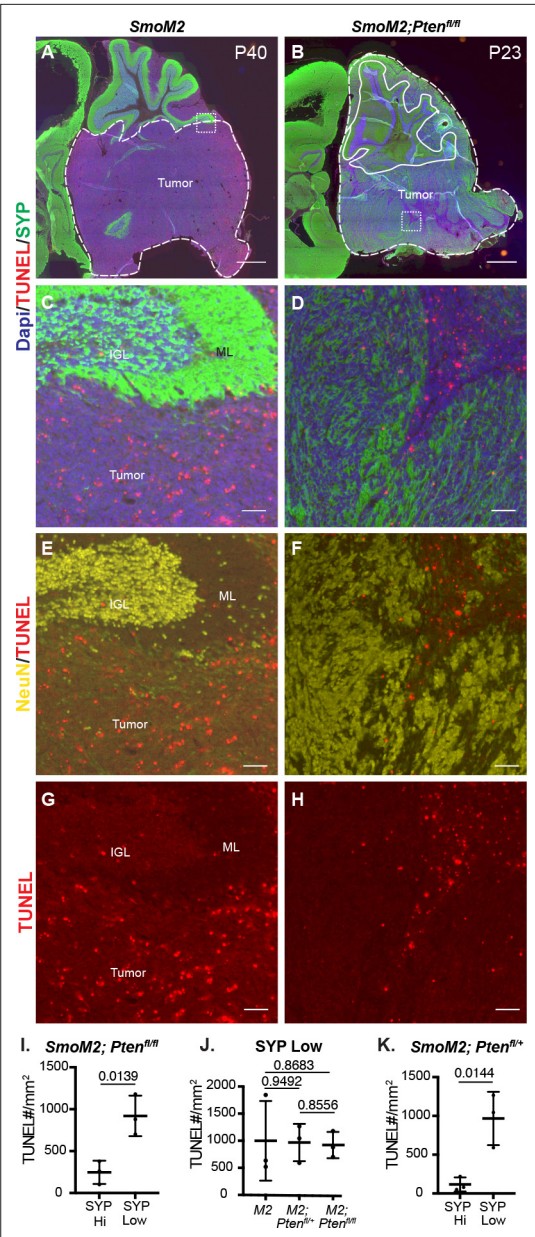

**Figure 3.** *Pten* loss in SmoM2 SHH-MB protects differentiated tumor cells from cell death. (**A, B**) Sagittal sections of end stage tumors in SmoM2 and SmoM2-*Pten*$^{fl/fl}$ mice stained as indicated (SmoM2, n=3; SmoM2-*Pten*$^{fl/fl}$ n=4). Dashed lines outline the tumors, solid lines outline IGL. Dotted line square indicates location of high-magnification images shown in (**C–H**). Scale bars indicate 1 mm. (**C–H**) High-magnification images of tumors stained as indicated showing reduced cell death (TUNEL particles) in differentiated regions (NeuN+ and SYP+) of tumors in SmoM2-*Pten*$^{fl/fl}$ mice compared to SmoM2 (SmoM2, n=3; SmoM2-*Pten*$^{fl/fl}$ n=4). Scale bars indicate 50 μm. (**I–K**) Quantification of density of TUNEL particles in tumor regions with high SYP (SYP HI) compared to low (SYP Low) staining in SmoM2-*Pten*$^{fl/fl}$ mice (**I**), in SYP Low regions in SmoM2 (M2), SmoM2-*Pten*$^{fl/+}$ (*M2; Pten*$^{fl/+}$) and SmoM2-*Pten*$^{fl/fl}$ (M2*; Pten*$^{fl/fl}$) mice (**J**), and in SYP High and Low regions of SmoM2-*Pten*$^{fl/+}$ mice (**K**). n=3 mice/genotype. All statistics were determined using an unpaired t-test comparing SYP High to Low (**I, K**) or two pairs of genotypes (**J**). Bars in scatter graphs represent mean ± SD for all the samples; individual data points are shown.

The online version of this article includes the following figure supplement(s) for figure 3:

**Figure supplement 1.** Method of quantifying TUNEL particles in tumors.

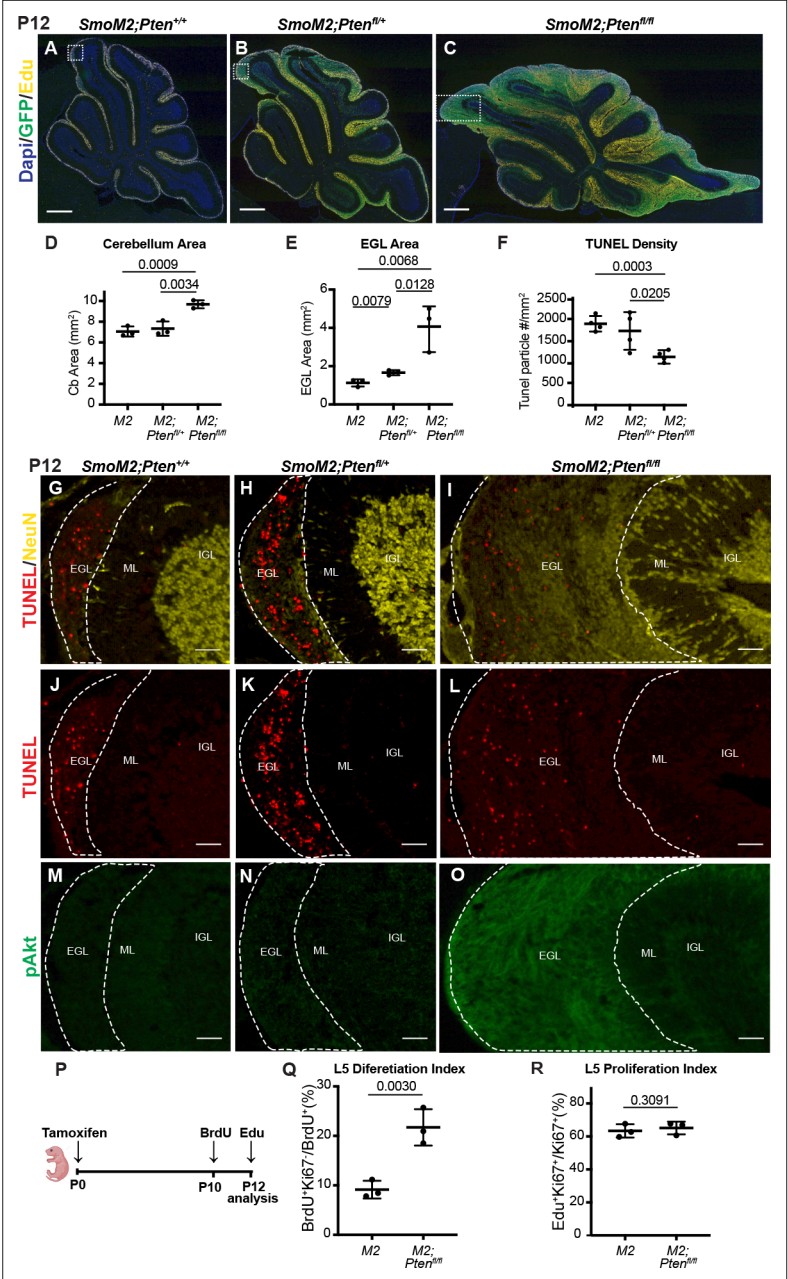

**Figure 4.** Loss of *Pten* leads to cell autonomous changes in SmoM2 GCPs at P12. (**A–C**) Midline cerebellar sagittal sections of P12 SmoM2, SmoM2-*Pten*^fl/+^, and SmoM2-*Pten*^fl/fl^ mice stained for GFP and EdU to show the EGL with proliferating mutant GCPs (SmoM2, n=11; SmoM2-*Pten*^fl/+^ n = 5; SmoM2-*Pten*^fl/fl^ n=6). Scale bars indicate 0.5 mm. Dotted line boxes indicate locations of images in (**A** for **G**, **J**, **M**; **B** for **H**, **K**, **N**; **C** for **I**, **L**, **O**). (**D**) Quantification of the area of the entire cerebellum on near midline sections of SmoM2 (M2), SmoM2-*Pten*^fl/+^ (M2; *Pten*^fl/+^), and SmoM2-*Pten*^fl/fl^ (M2; *Pten*^fl/fl^) mice (n=3 mice/genotype). (**E**) Quantification of the area of the entire EGL on near midline sections in the three genotypes (n=3 mice/genotype). (**F**) Quantification of TUNEL particle density (cell death) in the entire EGL of near midline sections in the three genotypes (n=3 mice/genotype) as described in *Figure 4—figure supplement 1*. (**G–O**) Staining of sections of the three genotypes for the proteins indicated showing reduced cell death (TUNEL density) and increased pAKT and NeuN in the EGL of SmoM2-*Pten*^fl/fl^ mice (n>3 per genotype). Scale bars indicate 50 µm. EGL, external granule layer; ML, molecular layer (region between IGL and EGL); IGL, internal granule layer. (**P**) Schematic of experimental design. (**Q**) Quantification of the percentage of proliferating GCPs that became post-mitotic (Ki67-) between BrdU labeling at P10 and analysis at P12 in the base of lobule 4/5 (L5) near the midline (n=3 mice/genotype and >300 cells/animal). (**R**) Quantification of the percentage of GCPs that are in S phase (EdU + Ki67+of all Ki67+GCPs) at P12 in the base of lobule 4/5

*Figure 4 continued on next page*

*Figure 4 continued*

(L5) near the midline (n=3 mice/genotype). All statistics were determined using an unpaired t-test comparing two pairs of genotypes. Bars in scatter graphs represent mean ± SD for all the samples; individual data points are shown.

The online version of this article includes the following figure supplement(s) for figure 4:

**Figure supplement 1.** Method of quantifying TUNEL particles in P12 EGL.

---

(*Figure 5A–E*). Quantification of the proliferation rate %GFP + and EdU + cells/total GFP + cells in the outer EGL (oEGL) of lobule 4/5 (L5) containing proliferating cells (region containing EdU + cells) was slightly but significantly higher in the SmoM2-*Pten*^*fl/fl*^ and the SmoM2-*Pten*^*fl/+*^ mice compared to SmoM2 (*Figure 5F*). Moreover, the percentage of *Pten* mutant/control (GFP+) cells that remained in the oEGL of all the GFP + cells in L5 (including those that differentiated) was significantly higher in the two *Pten* mutant genotypes (*Figure 5G*). This result indicates that loss or reduction of *Pten* initially induces a pro-proliferative or progenitor state in SmoM2-expressing GCPs. By P12, however, SmoM2-*Pten*^*fl/fl*^ GCP-lineage cells accumulate more in the EGL than do SmoM2-*Pten*^*fl/+*^ or SmoM2 GCPs, in part because there is an accumulation of NeuN+/SYP +granule cell (GC)-like cells, likely due to a survival advantage of these cells. This survival advantage suggests that the rapid growth of tumors in SmoM2-*Pten*^*fl/fl*^ mice is due to accumulation of GC-like cells and would account for the highly differentiated cytology of end-stage tumors.

## Loss of *Pten* in normal GCPs causes increased proliferation and decreased differentiation

Given that *Pten* loss initially increases proliferation and a progenitor state of SmoM2 (SHH activated) GCPs, we asked whether *Pten* loss in normal GCPs has the same effect. Using our mosaic MASTR

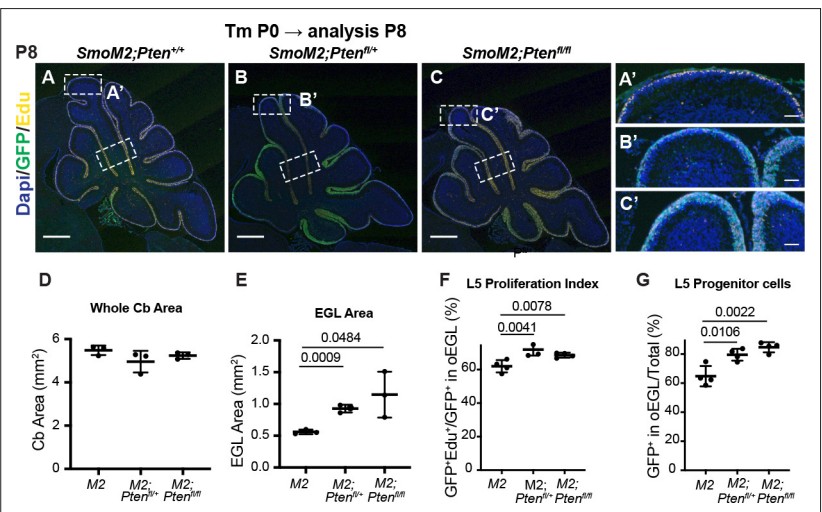

**Figure 5.** Loss of *Pten* leads to cell autonomous changes in SmoM2 GCP behaviors at P8. (**A–C**) Midline cerebellar sagittal sections of P8 SmoM2, SmoM2-*Pten*^*fl/+*^, and SmoM2-*Pten*^*fl/fl*^ mice stained for GFP and EdU to show the EGL with proliferating mutant GCPs. Dotted line boxes indicate locations of where quantifications were performed. Scale bars indicate 0.5 mm. (**A'-C'**) High-magnification images of areas indicated in (**A–C**). Scale bars indicate 0.5 µm. (**D**) Quantification of the entire (whole) area of the cerebellum on midline sections of SmoM2 (M2), SmoM2-*Pten*^*fl/+*^ (*M2; Pten*^*fl/+*^), and SmoM2-*Pten*^*fl/fl*^ (M2; *Pten*^*fl/fl*^) mice (n=3 mice/genotype). (**E**) Quantification of the area of the external granule layer (EGL) on entire midline sections in the three genotypes (n=3 mice/genotype). (**F**) Quantification of the proliferation index in the regions of lobule 4/5 (L5) shown in A-C at P8: percentage of mutant GCPs that are in S phase (GFP + EdU + of all GFP + GCPs in the outer EGL (oEGL) or region containing EdU + cells) (n=4 mice/genotype). (**G**) Quantification of the percentage of *Pten* mutant or control GCPs (GFP+) in lower region shown in A-C that remained in the proliferative oEGL at P8 and thus were progenitors. Analysis in F,G performed in wall of lobule 5 (lower rectangles in **A–C**) (n=4 mice/genotype). All statistics were determined using an unpaired t-test comparing two pairs of genotypes. Bars in scatter graphs represent mean ± SD for all the samples; individual data points are shown.

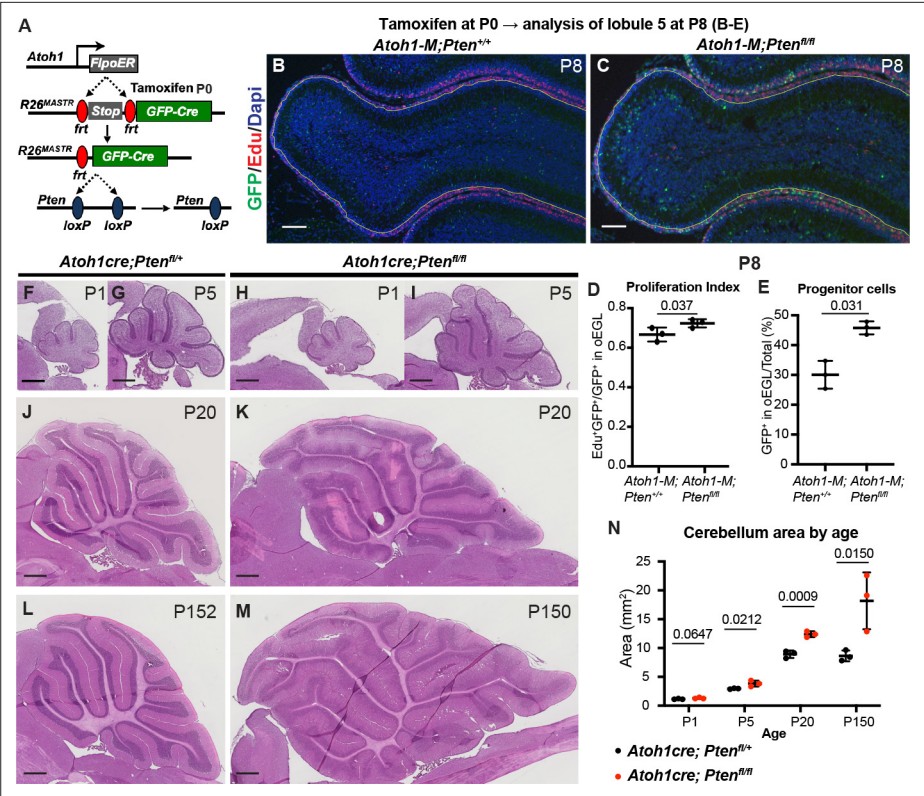

**Figure 6.** Loss of *Pten* in normal GCPs causes increased proliferation and decreased differentiation. (**A**) Schematic showing mosaic mutant analysis approach to study scattered GFP+ *Pten* homozygous mutant GCPs in *Atoh1-FlpoER/+; R26^{FSF-GFPcre/+}; Pten^{fl/fl}* (*Atoh1-M-Pten^{fl/fl}*) mice compared to control GFP+ cells in *Atoh1-FlpoER/+; R26^{FSF-GFPcre/+}; Pten^{+/+}* mice (*Atoh1-M- Pten^{+/+}*) at P8 following tamoxifen at P0. (**B, C**) Midline sagittal sections of lobule 4/5 stained with GFP marking mutant cells and EdU (S phase) at P8 in the two genotypes. Scale bars indicate 100 μm. (**D**) Quantification of the proliferation index in the entire lobule 4/5 (L5) at P8: percentage of *Pten* mutant/control GCPs that are in S phase (GFP+ and EdU+ of all GFP+ GCPs in the outer EGL (oEGL) defined by the region containing EdU+ cells; n=3 mice per genotype). (**E**) Quantification of the percentage of *Pten* mutant/control GFP+ GCPs in the entire L4/5 that remained in the proliferative oEGL at P8 and thus were progenitors and had not migrated inwards (n=3 mice per genotype). (**F–M**) Midline sagittal H&E stained sections of the cerebellum of conditional mutant mice lacking *Pten* in the GCP lineage (*Atoh1-Cre; Pten^{fl/fl}*) compared to controls (*Atoh1-Cre; Pten^{fl/+}*). Scale bars indicate 500 μm. (**N**) Quantification of the entire area of near midline cerebellar sagittal sections across development and ages (n=3 mice per age). All statistics used an unpaired t-test comparing two pairs of genotypes (**D, E**) or two genotypes per age (**N**). Bars in scatter graphs represent mean ± SD for all the samples; individual data points are shown.

approach, we used *Atoh1-FlpoER* to induce GFPcre in scattered GCPs at P0, with or without concomitant deletion of both floxed copies of *Pten* (*Figure 6A*). As in SmoM2 GCPs, we found at P8 a small but significant increase in the proliferation rate (percentage of EdU + cells in the proliferative oEGL defined by EdU +region) and in the proportion of GFP + cells that remained in the proliferative oEGL of *Pten* mutant GCPs (*Figure 6B–E*). Using *Atoh1-Cre* to delete *Pten* in all GCPs in the embryo (E14 onwards), we found that the cerebellar area of *Atoh1-Pten* conditional mutants (*Atoh1/+; Pten^{fl/fl}*) was significantly increased at P5 and later, but not at P1 compared to *Atoh1/+; Pten^{fl/+}* control mice (*Figure 6F–N*). Thus, PTEN normally constrains GCP proliferation and possibly differentiation.

## Loss of *Pten* in SmoM2 SHH-MB tumor cells results in a cell nonautonomous decrease in macrophage infiltration

Given the low density of dying cells in SmoM2-*Pten^{fl/fl}* SHH-MB, we asked whether the tumors have a change in the density of infiltrating macrophages. We previously showed that tumors in SmoM2 mice have a high infiltration of macrophages and that they are pro-tumorigenic in a majority of tumors,

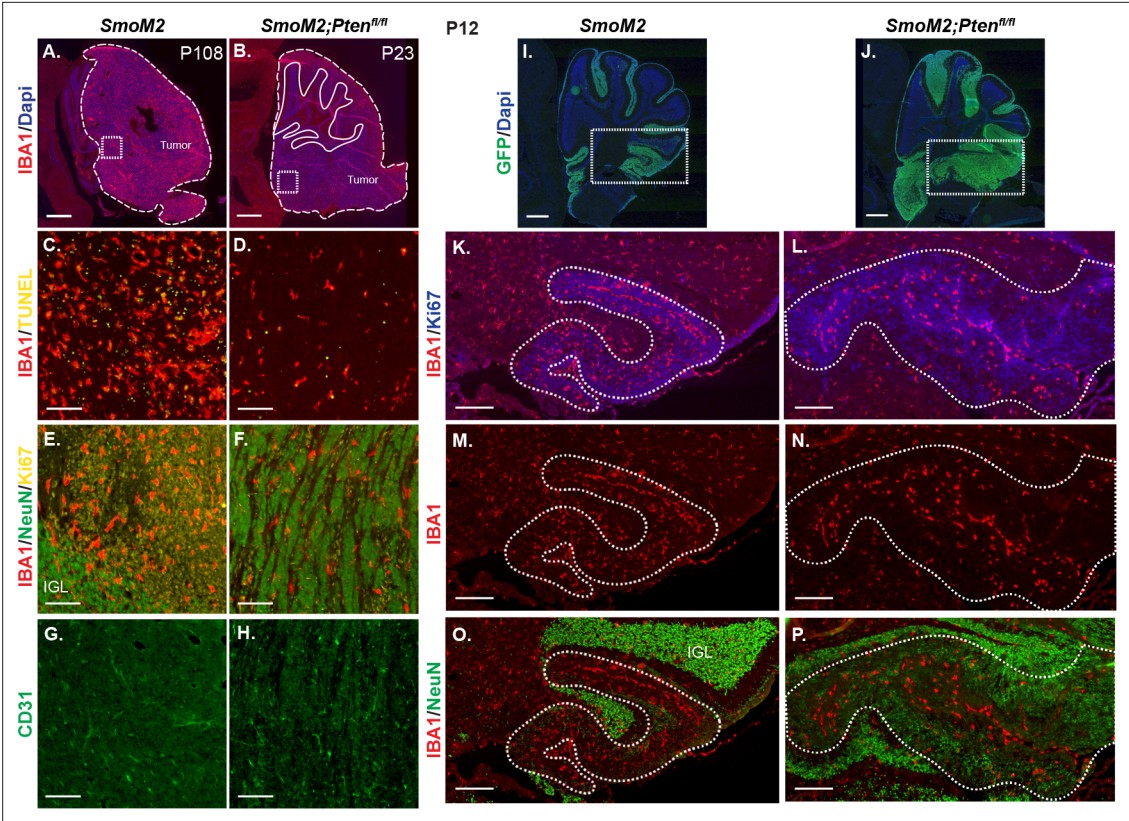

**Figure 7.** Loss of *Pten* in SmoM2 GCPs results in a cell non-autonomous decrease in macrophages in SHH-MB tumors. (**A, B**) Sagittal sections of end stage lateral tumors in SmoM2 and SmoM2-*Pten*^fl/fl^ mice stained with IBA1 (macrophage marker) and DAPI (n=3 mice/genotype). Dashed lines outline the tumors, solid lines outline IGL. Dotted line squares indicate locations of high-magnification images shown in (**C–H**). Scale bars indicate 1 mm. (**C–H**) Higher magnification images of tumors stained with IAB1 and TUNEL (**C, D**), IAB1, NeuN, and Ki67 showing enrichment of macrophages in undifferentiated regions of SmoM2-*Pten*^fl/fl^ mice (**E, F**) and with CD31 to label blood vessels (**G, H**) (n=5–9 mice/genotype). IGL, internal granule layer. Scale bars indicate 100 µm. (**I, J**) Lateral sagittal cerebellar sections of P12 SmoM2 and SmoM2-*Pten*^fl/fl^ mice stained for GFP and DAPI to show the EGL with mutant GCPs (n=3 mice/genotype). Dotted line boxes indicate locations of higher magnification images in (**K–P**). Scale bars indicate 500 µm. (**K–P**) Sections of P12 mice stained with IAB1, Ki67, and NeuN showing macrophages in large tumor lesions are enriched in proliferative regions of SmoM2-*Pten*^fl/fl^ mice (n=3 mice/genotype). Dotted line outlines the tumor lesion. Scale bars indicate 200 µm.

because reduction of macrophages using PLX5622 treatment leads to a decrease in tumor incidence (*Tan et al., 2021*). Staining of cerebellar tumor sections from SmoM2-*Pten*^fl/fl^ and SmoM2 mice at end stage revealed the expected high infiltration of IBA1 + macrophages in SmoM2 tumors, but also a striking decrease in macrophage density in tumors lacking *Pten* (*Figure 7A–D*). Moreover, co-staining with Ki67 or NeuN showed that in SmoM2-*Pten*^fl/fl^ mice, the macrophages were preferentially located in the areas with proliferating cells and thus increased cell death (*Figure 7C–F*). The decrease in macrophages in the differentiated regions of SmoM2-*Pten*^fl/fl^ mice is not due to a lack of blood vessels, based on CD31 staining of tumors (*Figure 7G and H*). At P12, in small lesions, the macrophages mainly accumulated in the meninges between the lobules, but in large lesions, they were more concentrated in regions of proliferation in both genotypes, and as in end stage, were decreased in NeuN + regions in SmoM2-*Pten*^fl/fl^ mice (*Figure 7I–P*). Thus, loss of *Pten* specifically in tumor cells leads to a cell nonautonomous decrease in macrophages in the tumor microenvironment from an early stage.

## scRNA-seq analysis reveals altered neural differentiation and suppression of immune signaling in tumor cells lacking *Pten*

To gain insight into transcriptional changes that occur in SmoM2 SHH-MB tumor cells that lack *Pten* and that could account for the cell autonomous changes seen in the tumors, we performed single cell RNA sequencing (scRNA-seq) of whole tumors dissected from SmoM2 and SmoM2-*Pten*^fl/fl^ mice (n=2 per genotype). We chose to analyze SHH-MBs from P16 SmoM2-*Pten*^fl/fl^ mice, since this is a few

days before mice show signs of disease, and waited until P22 for SmoM2 mice when the tumors are more visible. Marker analysis of sections of the tumors at the two ages confirmed that the cellular phenotypes of large tumors in SmoM2-*Pten*^fl/fl^ and SmoM2 mice were similar to tumors at end stage (*Figure 8—figure supplement 1*). The SmoM2-*Pten*^fl/fl^ tumors had extensive regions of neural differentiation that had reduced cell death (TUNEL particle density) and macrophage density (IBA1 + cells) compared to in proliferative regions (Ki67+) and in SmoM2 tumors at P22 or P16.

Tumor cells from the two genotypes separated well in a PCA plot after filtering out poor-quality cells and integrating the replicates (*Figure 8—figure supplement 2A–C*). Clustering of all cells (17,998 SmoM2 and 22,864 SmoM2-*Pten*^fl/fl^) was performed (*Hao et al., 2021*) to produce 20 clusters with most of the cells, as expected, expressing the GCP lineage marker *Pax6* (clusters 0–9 and 13), with nine additional small clusters that included astrocytes, oligodendrocytes, unipolar brush cells, macrophages, endothelial cells, and mesenchymal cells (pericytes and fibroblast-like cells) based on marker gene expression (*Figure 8A and B*, *Figure 8—figure supplement 2D and E*, *Figure 8—figure supplement 2—source data 1*). The cells from each sample were distributed in all clusters, although the proportions of cells from each genotype varied between clusters (*Figure 8C*, *Figure 8—figure supplement 2F and G*; *Figure 8—figure supplement 2—source data 2*). Within the *Pax6* +clusters 0–9, based on marker gene expression and cell cycle phase (*Figure 8D and E*), clusters 3, 4, and 8 were proliferative (Ki67+) and enriched for the GCP markers *Gli1*, *Atoh1*, and *Barhl1,* indicating highly proliferative GCP-like tumor cells. Cluster 6 had a mixture of cells in G2/M and G1, indicating cells transitioning from a proliferative to differentiated state. Clusters 1, 5, and 7 had post-mitotic (G1) cells that expressed early GC differentiation markers including *Barhl1* and *Slc17a6*, indicating early GC-like cells. Clusters 0, 2, and 9 were also in G1 but expressed the mature GC markers *Slc17a7* and *Syp* (*Figure 8E*) and had a low proportion of cells expressing *Gfp* (*Figure 8E*) and a much higher proportion of cells from SmoM2 mice than from SmoM2-*Pten*^fl/fl^ (*Figure 8C and F*, *Figure 8—figure supplement 2G*) indicating mature GC-like cells. Many of the GC-like cells (clusters 0, 2, 9) were likely wild type GCs from the IGL that would disproportionately contaminate the samples from SmoM2 mice as the tumors were smaller than in SmoM2-*Pten*^fl/fl^ mice. When only the GC-lineage clusters with extensive *Gfp* expression were considered (1, 3, 4, 5, 6, 7, 8), the proportions of cells from the two genotypes were fairly evenly distributed in all clusters (*Figure 8G*).

Pseudotime analysis (*Trapnell et al., 2014*; *Qiu et al., 2017*) reflecting the differentiation pathway of cells within clusters 0–9 with an origin in the GCP-like clusters was consistent with our cell type predictions, showing clusters 0, 2, and 9 are most mature and that the trajectory proceeds in order through the transitioning cluster 6, then through the early GC-like clusters 1 and 5 to the mature GC cluster 0 and on to cluster 2 or 9 (*Figure 8H*). The analysis further indicated that early GC-like cluster 7 has a separate differentiation pathway, possibly indicating that loss of PTEN suppresses full differentiation of the GC-like tumor cells. These results are also consistent with our IHC marker analysis of tumor sections that showed that SmoM2-*Pten*^fl/fl^ tumors have an abundance of GC-like cells expressing NeuN and SYP and further indicated such cells have an early GC-like state (clusters 1, 5, 7).

We next performed differential expression analyses (*Figure 9—figure supplement 1—source data 1*) between SmoM2 and SmoM2-*Pten*^fl/fl^ cells in the GCP-like tumor cell clusters (3, 4, 8) and found 1,011 genes were significantly upregulated and 3,710 were downregulated in the cells lacking *Pten* compared to SmoM2 cells (adjusted p-value≤0.05, *Figure 9A*, *Figure 9—source data 1*). Gene set enrichment analysis (GSEA) of Hallmark terms confirmed mTORC1 signaling is significantly upregulated in SmoM2-*Pten*^fl/fl^ tumors compared to SmoM2, as well as terms associated with MYC and E2F targets and unfolded protein response (*Figure 9B*, *Figure 9—figure supplement 1A*, *Figure 9—figure supplement 1—source data 1*). Violin plots of expression of mTORC1 genes supported upregulation of MTOR signaling (*Figure 9C*). As for pathways downregulated in SmoM2-*Pten*^fl/fl^ cells compared to SmoM2, 'Interferon alpha response' was greatest, and the list included 'complement' and 'inflammatory response' (*Figure 9D*), possibly contributing to the observed decrease in infiltrating macrophages in such tumors compared to SmoM2. Violin plots of genes involved in Interferon alpha signaling supported their downregulation in tumor cells lacking *Pten* (*Figure 9E*). GSEA of Gene ontology (GO) terms (biological processes) showed upregulation of terms associated with neural development in SmoM2-*Pten*^fl/fl^ tumors compared to SmoM2 (*Figure 9F*, *Figure 9—figure supplement 1B*, *Figure 9—figure supplement 1—source data 2*), indicating loss of PTEN pushes

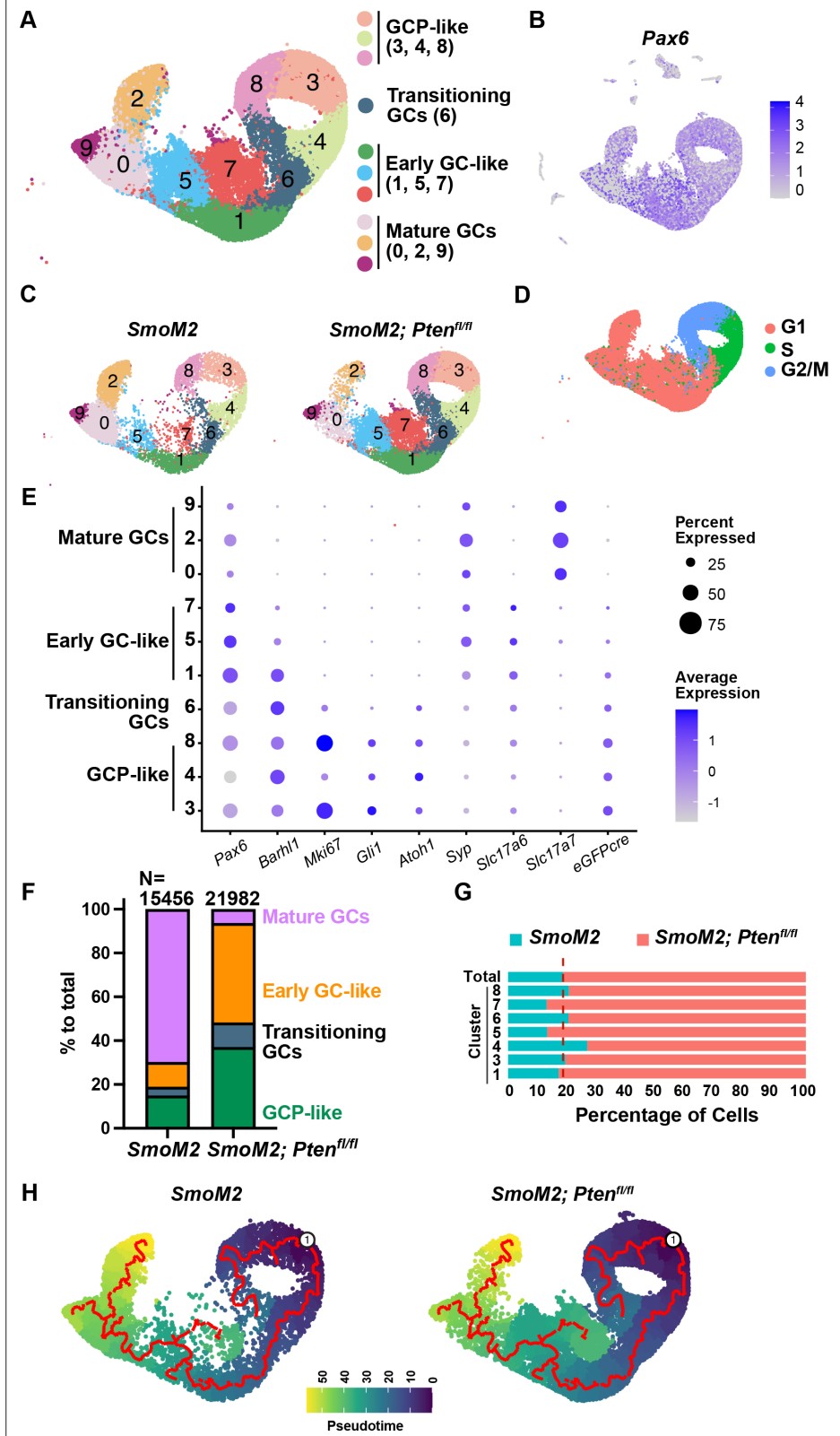

**Figure 8.** scRNA-seq analysis of tumors from SmoM2 and SmoM2-*Pten*<sup>fl/fl</sup> mice reveals altered differentiation in SHH-MB tumor cells lacking *Pten*. (**A**) UMAP of integrated cells from SmoM2 (n=2) and SmoM2-*Pten*<sup>fl/fl</sup> (n=2) tumors showing scRNA-seq GCP-lineage clusters 0–9 (*Pax6*-expressing) (left) and the assignment of cell types to each cluster (right) based on marker gene expression and cell cycle phase. (**B**) *Pax6* expression (GCP-lineage

*Figure 8 continued on next page*

*Figure 8 continued*

marker) shown across all clusters in UMAP of all tumor cells. (**C**) UMAP of GCP-lineage cells shown by genotype. (**D**) UMAP of GCP-lineage cells (two genotypes) showing cell cycle phases. (**E**) Dot plot graph showing expression levels of GCP-lineage marker genes and designation of the clusters as GCP-like or maturing GC-like cell types. (**F**) Graph of GCP-lineage cells showing percentage of cells from each genotype that are GCP-like, transitioning GC-like, early GC-like, and mature GC-like showing a higher percentage of mature GCs in tumors from SmoM2 mice. (**G**) Graph showing proportions of cells of each genotype in each of the GCP-lineage cell types excluding mature GCs that are mostly from SmoM2 mice. The dotted line represents the expected ratio between the two genotypes. (**H**) Pseudotime analysis reflecting the differentiation pathway of cells within the GCP-lineage with an origin in the GCP-like clusters, indicating cluster 7 is stalled or altered during differentiation.

The online version of this article includes the following source data and figure supplement(s) for figure 8:

**Figure supplement 1.** The cellular phenotypes of large tumors from P16 SmoM2-*Pten*^fl/fl^ mice and P22 SmoM2 mice are similar to end stage tumors.

**Figure supplement 2.** ScRNA-seq comparison of tumors from P16 SmoM2-*Pten*^fl/fl^ mice to P22 SmoM2 mice.

**Figure supplement 2—source data 1.** Top cell markers for each cluster in merged scRNA-seq data.

**Figure supplement 2—source data 2.** Cell numbers per cluster for each genotype based on merged scRNA-seq data.

GCP-like cells toward neural differentiation. Violin plots of expression of genes involved in neural regulation supported upregulation of neurogenesis (*Figure 9G*).

## scRNA-seq analysis confirms reduced cell death in differentiated cells of SmoM2-*Pten*^fl/fl^ tumors

Since our immunohistochemical analysis of SmoM2-*Pten*^fl/fl^ tumors showed that cell death was reduced in the NeuN+/SYP +regions of tumors, we next performed differential expression analyses of only the *Pten* mutant cells in the early GC-like cluster 1 to the GCP-like clusters 3, 4, and 8 and found 2738 genes were significantly upregulated and 4185 were downregulated in cluster 1 cells compared to GCP-like cells (adjusted p-value ≤0.05, *Figure 9H*, *Figure 9—source data 2*). GSEA of GO terms (biological processes) confirmed upregulation of terms associated with neural differentiation and also highlighted downregulation of 'positive regulation of programmed cell death' in *Pten* mutant cells in cluster 1 compared to the GCP-like clusters (*Figure 9I and K*, *Figure 9—figure supplement 1C*, *Figure 9—figure supplement 1—source data 3*). Violin plots of genes underlying the terms supported their altered expression in SmoM2-*Pten*^fl/fl^ tumors compared to SmoM2 (*Figure 9J and L*).

RNA in situ analysis of sections of P22 SmoM2 and P16 SmoM2-*Pten*^fl/fl^ tumors was used to confirm some of the genes shown to be differentially expressed between the GCP-like cells in the two genotypes (*Figure 9M–R*). *Asns* (Unfolded protein gene), *Vldlr,* and *Cdkn1a* (mTORC1 pathway) showed upregulation in GCP-like tumor cells in P16 SmoM2-*Pten*^fl/fl^ compared to P22 SmoM2 tumors.

## scRNA-seq analysis indicates loss of *Pten* in SmoM2 SHH-MB tumor cells results in decreased expression of genes associated with cytotoxicity in macrophages

Comparison of the transcriptomes of macrophages in SmoM2-*Pten*^fl/fl^ and SmoM2 tumors indicated an anti-inflammatory phenotype in tumors lacking *Pten* because genes related to synaptic signaling and *Irf2* were decreased, and *Cd36*, a marker of protumoral M2-type macrophages, was upregulated in SmoM2-*Pten*^fl/fl^ tumors (*Figure 9—figure supplement 1D and E*, *Figure 9—figure supplement 1—source data 4 and 5*; *Qin et al., 2024*; *Yang et al., 2022*). The observed reduction in macrophage infiltration into SmoM2-*Pten*^fl/fl^ tumors compared to SmoM2 tumors and the apparent reduction in cytotoxicity might reflect an immunosuppressive tumor microenvironment that could contribute to the rapid growth of tumors lacking PTEN.

## Discussion

Our work delineates cellular and transcriptional changes that lead to greatly accelerated tumor growth when *Pten* is mosaically removed in GCPs that have activated SHH-signaling due to expression

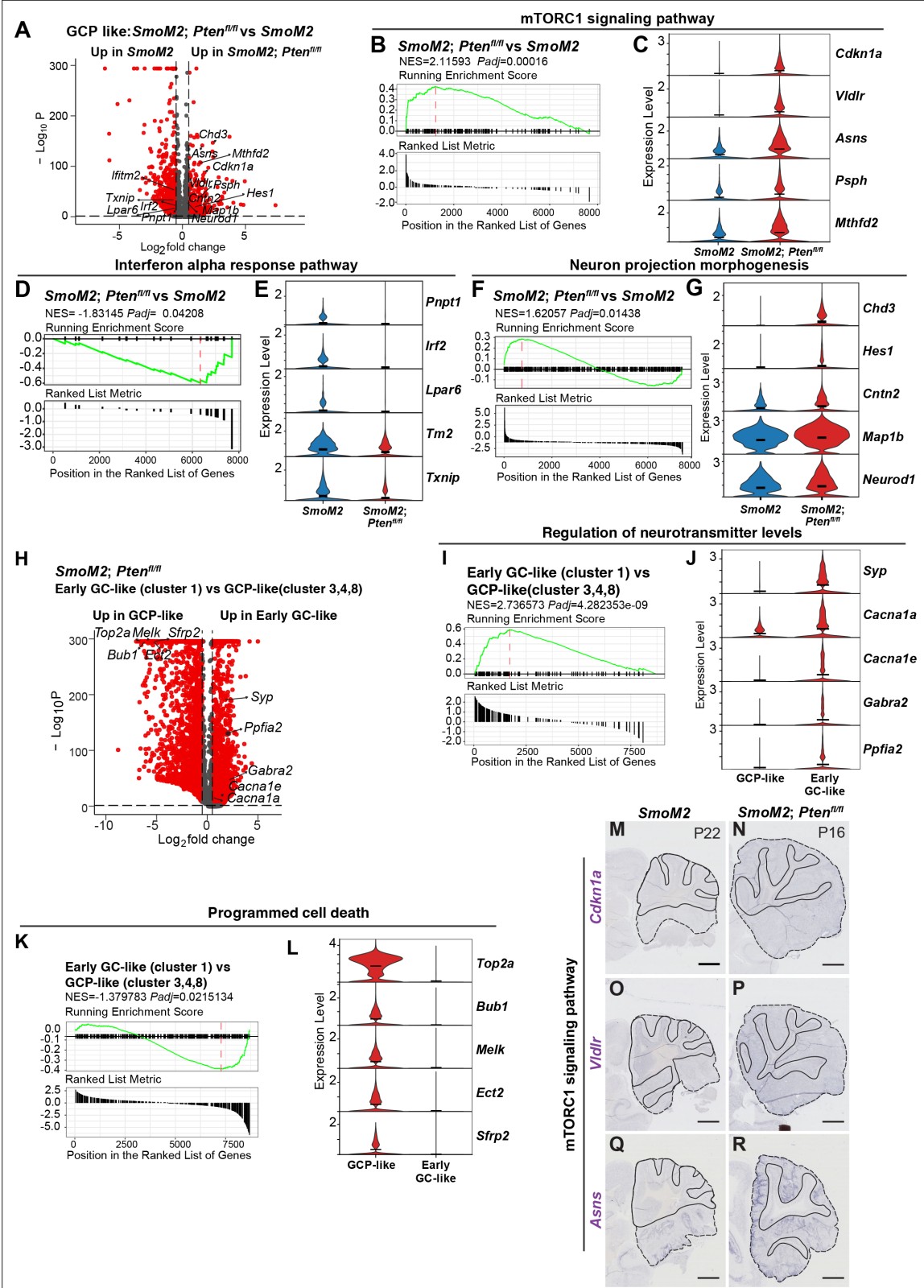

**Figure 9.** SmoM2 tumors lacking *Pten* have altered neurogenesis and immune signaling. (**A**) Volcano plot showing differentially expressed genes in GCP-like tumor cell clusters (3, 4, 8; red: adjusted p-value0.05, ab[log₂fold-change]>0.5), in SmoM2-*Pten*^fl/fl tumors compared to SmoM2. (**B**) Gene set enrichment analysis (GSEA) plots (Hallmark terms) showing that mTORC1 signaling pathway is significantly unregulated in SmoM2-*Pten*^fl/fl tumors compared to SmoM2. (**C**) Violin plots showing expression levels of mTORC1 signaling pathway genes in the two genotypes. (**D**) GSEA plots (Hallmark

*Figure 9 continued on next page*

*Figure 9 continued*

terms) showing that Interferon alpha response is significantly downregulated in SmoM2-*Pten*$^{fl/fl}$ tumors compared to SmoM2. (**E**) Violin plots showing expression levels of Interferon alpha response genes in the two genotypes. (**F**) GSEA plots of Gene ontology (GO) terms (biological processes) showing that regulation of neural differentiation is significantly upregulated in SmoM2-*Pten*$^{fl/fl}$ tumors compared to SmoM2. (**G**) Violin plots showing expression levels of neural differentiation genes in the two genotypes, indicating an early stage of neurogenesis is enhanced. (**H**) Volcano plot showing differentially expressed genes in early GC-like cluster 1 compared to GCP-like cell clusters (3, 4, 8) in SmoM2-*Pten*$^{fl/fl}$ tumors (red: adjusted p-value0.05, ab[log$_2$fold-change]>0.5). (**I**) GSEA plots of Gene ontology (GO) terms (biological processes) confirmed upregulation of neural differentiation in cluster 1 compared to the GCP-like cell clusters. (**J**) Violin plots showing expression levels of neural differentiation genes in the two cell types. (**K**) GSEA plots of Gene ontology (GO) terms (biological processes) showed downregulation of 'positive regulation of programmed cell death' in cluster 1 compared to the GCP-like cell clusters. (**L**) Violin plots showing expression levels of programmed cell death genes in the two cell types. (**M–R**) Lateral sagittal sections of tumors from SmoM2 mice (P22, M, O, Q) or SmoM2-*Pten*$^{fl/fl}$ mice (P16, N, P, R) stained for RNA in situ analysis of *Cdkn1a*, *Vldr*, *Asns*, indicating upregulation of mTORC1 signaling pathway in *Pten* mutant tumors. Note that *Asns* is also a gene involved in the unfolded protein response.

The online version of this article includes the following source data and figure supplement(s) for figure 9:

**Source data 1.** scRNA-seq differential gene expression analysis of GCP-like tumor cells comparing SmoM2; *Ptenfl/fl* to SmoM2.

**Source data 2.** scRNA-seq differential gene expression analysis in early GC-like (cluster 1) compared with GCP-like (clusters 3, 4, 8) in SmoM2; Ptenfl/fl tumors.

**Figure supplement 1.** Differential expression analyses confirm activation of mTOR signaling, altered neural differentiation, and decreased immune signaling following *Pten* loss.

**Figure supplement 1—source data 1.** GSEA analysis of Hallmark terms of DEG gene list of GCP-like cells in SmoM2; *Ptenfl/fl* compared with SmoM2 tumors.

**Figure supplement 1—source data 2.** GSEA analysis of Gene Ontology terms of DEG gene list of GCP-like cells in SmoM2; Ptenfl/fl compared with SmoM2 tumors.

**Figure supplement 1—source data 3.** GSEA analysis of Gene Ontology terms of DEG gene list in early GC-like (cluster 1) compared with GCP-like (clusters 3, 4, 8) of SmoM2; Ptenfl/fl tumors.

**Figure supplement 1—source data 4.** scRNA-seq differential gene expression analysis of macrophages in SmoM2; Ptenfl/fl compared with SmoM2 tumors.

**Figure supplement 1—source data 5.** GSEA analysis of Gene Ontology terms of DEG gene list of macrophages in SmoM2; *Ptenfl/fl* compared with SmoM2 tumors.

of SmoM2. We uncovered that in both wild type and SmoM2 GCPs, loss of *Pten* results in an initial increase in mitotic rate and likelihood to remain in the progenitor state. As SmoM2 tumor formation proceeds, tumors lacking *Pten* accumulate differentiated early GC-like cells that are stalled in their differentiation and are protected from cell death. Thus, the bulk of the *Pten* mutant tumors are post-mitotic cells. Macrophages seem to avoid or do not readily penetrate the differentiated regions of the tumors, perhaps because there are fewer dying cells. With possible relevance to human tumors, unlike previous models of *Pten* heterozygous mutant mice that model Cowden Syndrome or hamar-toma syndromes (**Blumenthal and Dennis, 2008**) or ones that involve embryonic conditional loss of *Pten* in all GCPs (**Castellino et al., 2010**; **Metcalfe et al., 2013**), we found that heterozygous loss of *Pten* in sporadic GCPs does not result in more aggressive tumors than activating SHH-signaling alone in rare GCPs. Since our model more closely reflects the sporadic nature of human tumor mutations, our results predict that heterozygous loss of *PTEN* will primarily have a negative impact on disease outcome when the mutations are germline (*PTEN*$^{+/-}$). If human *PTEN* mutant SHH-MB has a similar differentiated cellular phenotype to mouse models, then targeting such tumors with drugs to kill proliferating cells will not remove the bulk of the tumor and will require the drugs having access to the rare proliferating GCP-like tumor stem-like cells.

We found that loss or reduction of *Pten* in SmoM2 mice initially increases expansion of proliferating GCPs through increased proliferation and decreased differentiation such that by P8 (one week after gene mutation at P0) the EGL is expanded in SmoM2-*Pten*$^{fl/+}$ and SmoM2-*Pten*$^{fl/fl}$ mice compared to SmoM2. By P12, however, the EGL expansion is greatly enhanced only in SmoM2-*Pten*$^{fl/fl}$ mice, and this is accompanied by an abundance of differentiated cells and a significant reduction in cell death. Our label retention assay involving BrdU injection at P10 revealed that a greater proportion of BrdU$^+$ SmoM2-*Pten*$^{fl/fl}$ GCPs were Ki67$^-$ at P12 than in SmoM2 mice. This result indicates either that by P10 the loss of *Pten* results in an increase in differentiation of GCPs compared to in SmoM2 mice, which would be consistent with our scRNA-seq analysis, but also likely due to a greater survival of the differentiated cells since the density of TUNEL$^+$ particles is reduced in the EGL of SmoM2-*Pten*$^{fl/}$

*fl* mice at P12. Consistent with this, it was shown that NeuN + cells in SHH-MB tumors preferentially die compared to proliferating or quiescent tumor cells in a *Ptch1* mutant model (*Vanner et al., 2014*). This previous result is the opposite of what we observed in late-stage SmoM2-*Pten*$^{fl/fl}$ tumors where the proliferative compartments of the tumors had greater cell death than the NeuN$^+$ or SYP$^+$ regions and a similar density of TUNEL particles to in SmoM2 tumors. Thus, a key reason for the accelerated tumor growth in *Pten* mutant SHH-MB models is the survival of the differentiated compartment.

In a mouse *Neurod2*-SmoA1 model of SHH-MB, *Pten* heterozygosity leads to earlier onset of disease and a differentiated/nodular cellular phenotype when *Pten* is deleted in the germline or conditionally throughout the brain (*Nestin-Cre*) *Castellino et al., 2010*. Heterozygous loss of *Pten* in GCPs using an *Atoh1-Cre* transgene to delete *Pten*$^{fl/+}$ and *Ptch1*$^{fl/fl}$ apparently also slightly increases the time of onset and produces a differentiated phenotype (*Metcalfe et al., 2013*). In contrast, in our sporadic SHH-MB model (SmoM2-*Pten*$^{fl/+}$) where only rare GCPs lack one copy of *Pten* (and express SmoM2), the time of onset and penetrance are not accelerated and the tumors have a classic histology. Only in rare SmoM2-*Pten*$^{fl/+}$ clones of cells in a minority of tumors where PTEN protein is lacking, likely due to loss of heterozygosity, did we observe a differentiated phenotype with reduced cell death as seen in SmoM2-*Pten*$^{fl/fl}$ mice. The difference in the outcome of our model and the other two is either because of the SHH-activating mutations (SmoM2 vs SmoA1 expression or *Ptch1* loss) or more likely because when all GCPs lack a copy of *Pten* and have activated SHH-signaling, the microenvironment provides an advantage for *Pten* heterozygous cells to form a differentiated tumor. It is also possible that another cell type targeted by the other two *Cre* transgenes (*Netstin* and *Atoh1*) supports tumor growth when *Pten* loss is heterozygous.

A previous study provided evidence that increased mTOR signaling in SHH-MB promotes spinal cord metastasis by expressing *Shh* in GCPs along with an activated form of AKT (*Wu et al., 2012*; *Mumert et al., 2012*). However, in our sporadic model and the two other models of *Pten* loss in mouse SHH-MB (*Castellino et al., 2010*; *Metcalfe et al., 2013*), spinal cord metastasis is not increased. Thus, it appears that the way in which the mTOR pathway is activated affects metastasis. Perhaps related to this, in a model of *Trp53* heterozygous loss and *Rictor* homozygous loss, which reduces pAKT, GCPs were found to persist longer than normal in the EGL (*Akgül et al., 2018*), which is the same as what we found when *Pten* was deleted. Thus, the way the pathway downstream of PTEN is altered dictates the subsequent behavior of the mutant GCPs.

In our previous study, we showed that in *Kmt2d* mutant SHH-MB models, no difference in tumor size was detected at P12 but by P21 tumors lacking one or two copies of *Kmt2d* were more than twice the size of those with intact *Kmt2d* and had a classic cytoarchitecture. Thus, there is a delayed positive effect of reducing/ablating this chromatin modifier gene on tumor growth rate compared to loss of *Pten*, where there is a rapid expansion of the EGL. Another major difference between secondary loss of the two tumor suppressors in SHH-MB is that *Kmt2d* mutant tumors have nearly fully penetrant spinal cord metastasis. Thus, continued mechanistic studies in mouse models of recurrent mutations seen in SHH-MB should provide valuable insights into how the disease is expected to progress and provide ideas for the development of targeted therapies based on the cellular phenotypes of tumors not just the signaling pathways altered.

# Materials and methods

## Key resources table

| Reagent type (species) or resource | Designation | Source or reference | Identifiers | Additional information |
|---|---|---|---|---|
| Antibody | Rat monoclonal GFP | Nacalai Tesque | Cat# 440484 | IF (1:500 dilution) |
| Antibody | Rabbit polyclonal GFP | Invitrogen | Cat# A-11122 | IF (1:1000) |
| Antibody | Chicken polyclonal GFP | Aves Lab | Cat# GFP-1010 | IF (1:1000) |
| Antibody | Rabbit monoclonal Pten (D4.3) XP | Cell Signalling Tech | Cat# 9188 | IF (1:300) |

*Continued on next page*

*Continued*

| Reagent type (species) or resource | Designation | Source or reference | Identifiers | Additional information |
|---|---|---|---|---|
| Antibody | Rabbit monoclonal Pten | Cell Signalling Tech | Cat# 9559 | IF (1:200) |
| Antibody | Rabbit monoclonal pAKT (Ser473) | Cell Signalling Tech | Cat# 4060 | 1IF (:400) |
| Antibody | Mouse monoclonal P27 | BD Pharmingen | Cat# 610241 | IF (1:500), Antigen Retrieval |
| Antibody | Mouse monoclonal Calbindin D-28K | Swant | Cat# 300 | IF (1:1000) |
| Antibody | Mouse monoclonal NeuN | Millipore (Chemicon) | Cat# MAB377 | IF (1:500), Antigen Retrieval |
| Antibody | Guinea pig polyclonal NeuN | Millipore (Chemicon) | Cat# ABN90 | IF (1:1000) |
| Antibody | Rabbit monoclonal Ki67 (clone SP6) | Fisher Scientific | Cat# RM-9106-S0 | IF (1:1000) |
| Antibody | Mouse monoclonal Ki67 | BD Pharmingen | Cat# 556003 | IF (1:1000), Antigen Retrieval |
| Antibody | Rabbit monoclonal SYP | Abcam Inc. | Cat# ab32127 | IF (1:500) |
| Antibody | Rabbit polyclonal pS6 (Ser235/236) | Cell Signalling Tech | Cat# 2211 | IF (1:200) |
| Antibody | Rat monoclonal BrdU | Abcam Inc. | Cat# ab6326 | IF (1:500), Antigen Retrieval |
| Antibody | Rabbit polyclonal IBA1 antibody | Wako Chemicals USA | Cat# 019–19741 | IF (1:1000) |
| Antibody | Rat monoclonal CD31 | BD Biosciences | Cat# 550274 | IF (1:500) |
| Commercial assay or kit | Click-iT EdU Cell Proliferation Kit | Invitrogen | Cat# C10340 | |
| Commercial assay or kit | Click-it EdU assay with Sulfo-Cyanine3 azide | Lumiprobe Corporation | Cat# A3330 | 0.2:1000 dilution |
| Commercial assay or kit | Click-it EdU assay with Sulfo-Cyanine5 azide | Lumiprobe Corporation | Cat# A1330 | 0.2:1000 |
| Commercial assay or kit | TdT enzyme for tunel assay | Roche | Cat# 3333574001 | IF (1:500) |
| Commercial assay or kit | EasySep Dead Cell Removal | StemCell | Cat# 17899 | |
| Commercial assay or kit | DIG RNA labeling | Sigma-Aldrich | Cat# 11277073910 | |
| Chemical compound, drug | Biotin-16-dUTP | Sigma-Aldrich | Cat# 11093070910 | 1:500 dilution |
| Chemical compound, drug | BM Purple AP Substrate | Sigma-Aldrich | Cat# 11442074001 | |
| Chemical compound, drug | Streptavidin Alexa Fluor 647 Conjugate | Invitrogen | Cat# S-32357 | 1:1000 |

*Continued on next page*

*Continued*

| Reagent type (species) or resource | Designation | Source or reference | Identifiers | Additional information |
|---|---|---|---|---|
| Chemical compound, drug | Digoxigenin-dUTP, alkali-stable | Enzo Life Sciences, Inc. | Cat# ENZ-NUC113-0025 | 1:500 |
| Chemical compound, drug | Sheep polyclonal Anti-Digoxigenin-Rhodamine, Fab fragment | Sigma-Aldrich | Cat# 11207750910 | 1:200 |
| Chemical compound, drug | Alexa Fluor 647 IgG Fraction | Jackson Immuno Research | Cat# 200-602-156 | 1:200 |
| Chemical compound, drug | Tamoxifen | Sigma Aldrich | Cat# T5648-1G | |
| Chemical compound, drug | Digitonin | Promega | Cat# G9441 | |
| Chemical compound, drug | EdU | Invitrogen | Cat# E10187 | |
| Chemical compound, drug | BrdU | Sigma-Aldrich | Cat# B9285-250MG | |
| Chemical compound, drug | fluoromount-g | Electron Microscopy Sciences | Cat# 17984–25 | |
| Chemical compound, drug | Protease/phosphatase inhibitor | Thermo Fisher Scientific | Cat# 78440 | |
| Chemical compound, drug | Papain Dissociation | Worthington | Cat# LK003150 | |
| Recombinant DNA reagent | pBS-Syp in Bluescript vector | BioMatik pBS-Syp | | |
| Recombinant DNA reagent | Syp Forward | BioMatik pBS-Syp | PCR primers | AAGATGGCCACTGACCCA |
| Recombinant DNA reagent | Syp Reverse | BioMatik pBS-Syp | PCR primers | AGGGCTGGGGAACCGATAGG |
| Recombinant DNA reagent | Asns FT3: | Sigma-Aldrich | PCR primers | GAATCGATTAACCCTCACTAAAGGCTATGCTGGACGGGGTGTT |
| Recombinant DNA reagent | Asns RSp6: | Sigma-Aldrich | PCR primers | CCCGGGATTTAGGTGACACTATAGGCCTCCTCCTCGGCCTTCTC |
| Recombinant DNA reagent | Cdkn1a FT3: | Sigma-Aldrich | PCR primers | GAATCGATTAACCCTCACTAAAGGCAGGCACCATGTCCAATCCTG |
| Recombinant DNA reagent | Cdkn1a RSp6: | Sigma-Aldrich | PCR primers | CCCGGGATTTAGGTGACACTATAGGCGAGCTTGGGTTGGGAGGGGC |
| Recombinant DNA reagent | Vldlr FT3: | Sigma-Aldrich | PCR primers | GAATCGATTAACCCTCACTAAAGGCGTATCCACGGCAGCAGGC |
| Recombinant DNA reagent | Vldlr RSp6: | Sigma-Aldrich | PCR primers | CCCGGGATTTAGGTGACACTATAGGCCACAGCTATGGAGGCAGGC |

*Continued on next page*

*Continued*

| Reagent type (species) or resource | Designation | Source or reference | Identifiers | Additional information |
|---|---|---|---|---|
| Strain, strain background (*M. musculus*) | *Atoh1-FlpoER/+* | Jackson Laboratory | Cat# 040091 | Ref 41 |
| Strain, strain background (*M. musculus*) | *R26*<sup>FSF-eGFP-Cre</sup> | Jackson Laboratory | Cat# 018903 | Ref 22 |
| Strain, strain background (*M. musculus*) | *SmoM2* | Jackson Laboratory | Cat# 005130 | Ref 24 |
| Strain, strain background (*M. musculus*) | *Pten*<sup>fl/fl</sup> | | | Ref 25 |
| Strain, strain background (*M. musculus*) | *Atoh1-Cre* | Jackson Laboratory | Cat# 011104 | Ref 42 |

## Animals

The animal care and procedures were performed in accordance with the Memorial Sloan Kettering Cancer Center Institutional Animal Care and Use Committee guidelines (IACUC protocol: 07-01-001). Mice were kept on a 12/12 hr light/dark cycle and in temperature- and humidity-controlled rooms with ad libitum access to standard laboratory mouse chow and water. All transgenic mouse lines were maintained on a mixed genetic background containing 129, C57BL/6 J, and Swiss Webster. Both sexes were randomly used for the analysis and experimenters were blinded for genotypes whenever possible. The following mouse lines were used in the study: *Atoh1-Cre* (JAX #011104; *Matei et al., 2005*), *R26*<sup>LSL-eGFPcre</sup> (MASTR, JAX #018903; *Lao et al., 2012*), *Atoh1-FlpoER* (JAX #040091; *Tan et al., 2018*; *Wojcinski et al., 2019*), *Pten*<sup>fl/fl25</sup> and *R26*<sup>LSL-SmoM2</sup> (SmoM2, JAX#005130; *Mao et al., 2006*). Genetic recombination was induced with FlpoER by injecting one 100 µg/g dose of Tamoxifen (Tm; Sigma-Aldrich) subcutaneously into the back of P0 mice. Tm was prepared in corn oil (Sigma-Aldrich) at 20 mg/mL and stored at 4 °C. All analyses were on a minimum of three mice of each genotype, the numbers are stated in the figure legends. No statistical methods were used to predetermine sample sizes, but our sample sizes are similar to those reported previously.

## Tissue preparation

For all histological analyses, mice were anesthetized with isoflurane and then perfused transcardially with room temperature (RT) phosphate buffered saline (PBS) with heparin (0.02 mg/mL) and then ice cold 4% paraformaldehyde (PFA, Electron Microscopy Sciences, catalog no: 15714). Brains and spinal cords were dissected and post-fixed in 4% PFA 3–4 days at 4 °C and cryopreserved in 30% sucrose in PBS for ~2 days at 4 °C. Brains were embedded in Tissue-Tek OCT compound (Sakura Finetek). Brains were serially cryosectioned sagittally from one side of the brain to the other for P16 and P22 brains and to the midline for end stage brains at 12 µm and collected on charged glass slides (Fisherbrand ColorFrost Plus) and stored at –20 °C. Frozen spinal cords were coronally sectioned at 20 um. Details of reagents are listed in the Key Resources Table.

## Hematoxylin and eosin (H&E) staining

H&E staining was obtained from Richard-Allan Scientific and included hematoxylin 2 solution, eosin-Y, bluing reagent, and clarifier 2. Slides were rinsed in PBS for 5 min and incubated in hematoxylin 2 solution for 3 min and then rinsed in deionized water (diH$_2$O) and placed in staining reagents for 1 min each (diH$_2$O, clarifier 2, diH$_2$O, bluing reagent, diH$_2$O, eosin-Y). Dehydration and defatting were then performed (dH$_2$O-70% ethanol-95%–95%–100%–100%-xylene-xylene-xylene, 1 min each) and then slides were mounted with a coverslip and DPX mounting medium (Electron Microscopy Sciences). Details of reagents are listed in the Key Resources Table.

## Immunofluorescence

Slides were washed three times for 5 min in PBS. When necessary, antigen retrieval was then performed by immersing slides in sodium citrate buffer (10 mM sodium citrate with 0.05% Tween-20, pH 6.0) for 20 min at 95 °C followed by rinsing in PBS. Slides were then incubated in blocking buffer (5% BSA in 0.4% Triton-X100 in PBS [PBST]) for 1 hr at RT. Slides were then placed in primary antibody solution in blocking buffer at 4 °C overnight. Primary antibody information and dilutions are listed in Key Resources Tables. Slides were then rinsed three times in RT 0.1% PBST and incubated in secondary antibody solution in blocking buffer (1:500) at RT for 1 hr. Slides then were incubated in Hoechst (Invitrogen, catalog #H3569 diluted 1:1000 in PBS) for 10 min and rinsed three times in PBS and cover-slipped with Fluoro-Gel (Electron Microscopy Sciences).

## TUNEL staining

TUNEL staining was according to the manufacturers' instructions (Roche TUNEL assay kit). Tissue on slides was permeabilized with 0.5% Triton X-100 and then pre-incubated with 1 x TdT reaction buffer with 5 mM $CoCl_2$ and 1 X PBS for 15 min at RT. Slides were then incubated for 1 hr at 37 °C in 1 x TdT reaction buffer containing Terminal Transferase (Roche) and Biotin-16-dUTP (Sigma-Aldrich) or Digoxigenin-dUTP (Enzo Life Sciences, Inc). Slides were then incubated with a Streptavidin Alexa Fluor 647 conjugate (Invitrogen catalog #S-32357) or Anti-Digoxigenin-Rhodamine (Sigma-Aldrich catalog), respectively, for 1 hr.

## BrdU injection and staining

To assess cell differentiation, BrdU (Sigma-Aldrich) was injected intraperitoneally at 50 µg/g at P10 for analysis at P12. Tissues were collected and processed as described above. For immunostaining, antigen retrieval was applied to slides with Rat primary anti-BrdU antibody BU1/75 (ICR1) (Abcam Inc) to detect BrdU.

## EdU (5-ethynyl-2'-deoxyuridine) injection and staining

To assess cell proliferation, EdU (Invitrogen) was injected intraperitoneally at 50 µg/g 1 hr before euthanasia. A Click-it EdU assay was used (Invitrogen) with Sulfo-Cyanine5 azide (Lumiprobe Corporation) per the manufacturer's protocol to stain sections.

## RNA in situ hybridization

Sagittal sections (12 µm) were processed for in situ hybridization as described (*Blaess et al., 2011*). Except for *Gli1* and *Syp,* antisense riboprobes were labeled with UTP-digoxigenin using a DIG RNA labeling mix (Sigma-Aldrich) with PCR-amplified templates prepared from cDNA synthesized from an early postnatal mouse cerebellum lysate. Signal was detected with alkaline phosphatase-coupled anti-digoxigenin antibodies using BM purple (Sigma-Aldrich) as the substrate. The primers used for PCR amplification were based on the Allen Brain Atlas. *Gli1* RNA in situ was performed using an antisense RNA probe made from a plasmid (*Qin et al., 2024*) and for *Syp* BioMatik generated a plasmid in Bluescript containing *Syp* sequences using the primers described in the Key Resources Table. Primers were flanked in the 3' with SP6 (antisense) and 5' with T3 (sense) promoters. Primers for RNA in situ probes are listed in the Key Resources Table.

## Microscopy, image processing, and analyses

Images were acquired with a DM6000 Leica fluorescent microscope (Leica Camera, Wetzlar, Germany) or NanoZoomer Digital Pathology microscope (Hamamatsu Photonics, Shizuoka, Japan) and were processed and analyzed using Fiji (*Schindelin et al., 2012*) or Photoshop (Adobe Inc, San Jose, CA, USA). Cell counts were manually obtained using the Cell Counter plugin for Fiji or semi-automated using the Analyze Particle plugin for Fiji as previously described (*Willett et al., 2019*).

The same region of the brain was analyzed in each animal for each assay, and the regions are stated in the figure legends. For early postnatal stages, sections near the midline were analyzed. For analysis of tumors, they were done on lateral sections.

## ScRNA-seq sample preparation

*Atoh1-FlpoER/+; R26^MASTR/LSL-SmoM2;Pten^+/+* (SmoM2) and *Atoh1-FlpoER/+; R26^MASTR/LSL-SmoM2; Pten^fl/fl* (SmoM2 *Pten^fl/fl*) mice were euthanized at P16 and P22, respectively, and the tumors were dissected

from the lateral-posterior cerebellum in ice-cold Hank's balanced salt solution (HBSS, Gibco). Then the tumors were dissociated in Papain (Worthington) for 30 min at 37 °C before being washed out in Oval-bumin (Worthington). Tumors were triturated by pipetting up and down using a P1000 and filtered through a 40 μm cell strainer (Falcon). Once the cells were dissociated, the dead cells were removed using the kit EasySep Dead Cell Removal (Annexin V) (StemCell). Then cells were centrifuged at 500 × $g$ at 4 °C for 5 min. Cells were then washed with PBS. For every sample (2 SmoM2 and 2 SmoM2 $Pten^{fl/fl}$), cells were counted using 0.2% Trypan on hemocytometer and diluted before loading to 10 X chip, targeting 10,000 viable cells per sample.

The scRNA-Seq of tumor cell suspensions was performed on a Chromium instrument (10 X genomics) following the user guide manual for 3' v3.1. Final libraries were sequenced on an Illumina NovaSeq6000 by the IGO core of MSKCC.

## scRNA-seq data analysis

The Cell Ranger Single Cell software suite (10 x Genomics) was used to align reads and generate feature-barcode matrices. The reference genome used was the Genome Reference Consortium Mouse Build 38 (GRCm38, Gencode annotation mm10). Raw reads were processed using the Cell Ranger count program using default parameters.

Seurat v5.0.2 package was used to generate a UMI (unique molecular identifier) count matrix from the Cell Ranger output *Satija et al., 2015*. Genes expressed in less than 10 cells were removed for further analyses. Cells with larger than 18000 UMIs, 0.20 mitoRatio, UMI ≤500 and≤500 genes detected were considered low quality/outliers and discarded from the datasets. Normalization was performed on individual samples using the NormalizeData function with default parameters. The normalized data was scaled by the ScaleData function with mitochondrial gene percentage regressed out. Principal component analysis (PCA) was performed on the scaled data by runPCA. Samples were then integrated using IntegrateLayers function with the Harmony Integration method. The first 20 dimensions were used for the FindNeighbors function, and clusters were identified using the Find-Clusters function with a resolution of 0.5. Data were projected onto the 2D space using the FindUMAP or RunUMAP function with 20 dimensions. Cluster markers and further differential gene expression analyses were performed using normalized counts (NormalizeData) in the RNA assay. Cluster markers were identified using the FindAllMarkers and comparing markers generated to the literature. To refine clustering further, the SubsetData function was used to create a new Seurat object, and the above clustering was reiterated.

Differential gene expression analyses between genotypes or clusters were performed using the FindMarkers function in the Seurat package. Genes with an adjusted p value (*Padj*)<0.05 were considered significantly up or downregulated. Results were visualized by violin plot and UMAP plot using the Seurat package, and volcano plot using the EnhancedVolcano package on Bioconductor (*Blighe et al., 2025*). GSEA analysis on GO and Hallmark terms was performed using the enrichGO, cluster-Profiler packages on Bioconductor (*Yu et al., 2012*). To build a pseudotime trajectory using the UMAP data, the Seurat object was converted to a monocle 3 (1.3.7) object by as.cell_data_set function (*Trapnell et al., 2014*; *Qiu et al., 2017*). Cells were then clustered by the cluster cells function. The trajectory GCP-lineage partition was constructed using the learn graph function followed by the order cells function.

## Statistical analysis

All statistical analyses to quantify data were performed using Prism software (GraphPad) and significance was determined as p ≤ 0.05. Each statistical analysis used is described in the figure legends. Data (shown as scatter graphs) are presented as mean ± standard deviation.

## Acknowledgements

We thank members of the Joyner lab for interesting discussions pertaining to SHH-MB and technical support, in particular Reeti Mayur Sanghrajka for insightful suggestions at the beginning of the study. We thank Andy McMahon for the SmoM2 mice and the Memorial Sloan Kettering Cancer Center Flow Cytometry Core, Integrated Genomics Operations Core Facility, and the Center for Comparative Medicine and Pathology for technical support. NCI (R01CA192176), Alexandra L Joyner. MSK

Functional Genomics Initiative (GC242211), Alexandra L Joyner. NCI Cancer Center Support Grant (CCSG, P30 CA08748), Alexandra L Joyner. Cycle for Survival, Alexandra L Joyner.

## Additional information

### Funding

| Funder | Grant reference number | Author |
|---|---|---|
| National Cancer Institute | R01CA192176 | Alexandra L Joyner |
| National Cancer Center | CCSG,P30 CA08748 | Alexandra L Joyner |
| MSK Functional Genomics Initiative | GC242211 | Alexandra L Joyner |
| Cycle for Survival | | Alexandra L Joyner |

The funders had no role in study design, data collection and interpretation, or the decision to submit the work for publication.

### Author contributions

Zhimin Lao, Formal analysis, Investigation, Methodology, Writing – review and editing; Salsabiel El Nagar, Data curation, Investigation, Writing – review and editing; Yinwen Liang, Data curation, Formal analysis, Writing – review and editing; Daniel Stephen, Investigation, Writing – review and editing; Alexandra L Joyner, Conceptualization, Data curation, Supervision, Funding acquisition, Writing - original draft, Project administration, Writing – review and editing

### Author ORCIDs

Zhimin Lao ⓘ https://orcid.org/0009-0009-3104-7948
Salsabiel El Nagar ⓘ https://orcid.org/0000-0002-7015-0907
Yinwen Liang ⓘ https://orcid.org/0000-0002-9945-3768
Daniel Stephen ⓘ https://orcid.org/0009-0000-4199-4900
Alexandra L Joyner ⓘ https://orcid.org/0000-0001-7090-9605

### Ethics

This study was performed in strict accordance with the recommendations in the Guide for the Care and Use of Laboratory Animals of the National Institutes of Health. All of the animals were handled according to approved institutional animal care and use committee (IACUC) protocols (07-01-001) of MSKCC. For histological analyses, mice were anesthetized with isoflurane and then perfused transcardially with room temperature (RT) phosphate buffered saline (PBS) with heparin (0.02 mg/mL) and then ice cold 4% paraformaldehyde.

Reviewer #1 (Public review): https://doi.org/10.7554/eLife.108190.3.sa1
Reviewer #2 (Public review): https://doi.org/10.7554/eLife.108190.3.sa2
Author response https://doi.org/10.7554/eLife.108190.3.sa3

## Additional files

### Supplementary files

MDAR checklist

### Data availability

RNA sequence data were deposited in GEO (accession number GSE295733). All data generated or analyzed during this study are included in the manuscript and supporting files; source data files have been provided for all figures.

The following dataset was generated:

| Author(s) | Year | Dataset title | Dataset URL | Database and Identifier |
|---|---|---|---|---|
| Liang Y, Nagar S, Joyner A | 2025 | PTEN restrains tumor expansion through cell autonomous and non-autonomous mechanisms | https://www.ncbi.nlm.nih.gov/geo/query/acc.cgi?acc=GSE295733 | NCBI Gene Expression Omnibus, GSE295733 |

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
