## [Editor Report · eLife Assessment]

This **valuable** study provides insights into the role of Pten mutations in SHH-medulloblastoma, by using mouse models to resolve the effects of heterozygous vs homozygous mutations on proliferation and cell death throughout tumorigenesis. The experiments presented are **convincing**, with rigorous quantifications and orthogonal experimentation provided throughout, and the models employing sporadic oncogene induction, rather than EGL-wide genetic modifications, represent an advancement in experimental design. However, additional experimentation focused on a greater characterization of macrophage phenotypes (e.g., microglia vs circulating monocytes) would enhance this study. The work will be of interest to medical biologists studying general cancer mechanisms, as the function of Pten may be similar across tumor types.

---

## [Referee Report · Reviewer #1 (Public review)]

This study investigates how Pten loss influences medulloblastoma development in mouse models of Shh-driven MB. Previous studies have shown that Pten heterozygosity can accelerate tumorigenesis in models where the entire GNP compartment harbours MB-promoting mutations, raising questions about how Pten levels and context interact, especially when MB-initiating mutations occur sporadically in the cerebellum. Here, the authors create an allelic series combining sporadic, cell-autonomous induction of oncogenic SmoM2 with Pten loss in granule neuron progenitors. In contrast to previous studies, Pten heterozygosity does not significantly impact tumour development from sporadic SmoM2 induction, whereas complete Pten loss accelerates tumour onset. Analysis of Pten-deficient tumours reveals accumulation of death-resistant differentiated cells and reduced macrophage infiltration. At early stages, Pten-deficient pre-tumour cells exhibit increased proliferation and EGL hyperplasia, indicating that Pten loss drives proliferation but shifts cells towards differentiation.

Strengths

This study raises the bar for modelling and interpreting the effects of secondary mutations on MB development. It is carefully executed, and the models-using sporadic oncogene induction rather than EGL-wide genetic manipulations-represent an advance in experimental design. The deeper phenotyping, including single-cell RNA-seq and target validation, adds rigor. This work extends previous work on ShhMB and Pten by showing that Pten heterozygosity in GNPs is likely not responsible for the accelerated tumour development reported in earlier studies. The evolution of these Pten-deficient tumours from proliferative to post-mitotic and death-resistant is an important observation with potential clinical significance.

Minor weakness

The absence of an effect of Pten heterozygosity on tumour development in their model suggests non-cell-autonomous effects, but this is not directly demonstrated. Changes in macrophage recruitment warrant further exploration and represent an interesting avenue for future investigation.

---

## [Referee Report · Reviewer #2 (Public review)]

The authors sought to answer several questions about the role of the tumor suppressor PTEN in SHH-medulloblastoma formation. Namely, whether Pten loss increases metastasis, understanding why Pten loss accelerates tumor growth, and the effect of single-copy vs double-copy loss on tumorigenesis. Using an elegant mouse model, the authors found that Pten mutations do not increase metastasis in a SmoD2-driven SHH-medullolbastoma mouse model, based on extensive characterization of the presence of spinal cord metastases. Upon examining the cellular phenotype of Pten-null tumors in the cerebellum, the authors made the interesting and puzzling observation that Pten loss increased the differentiation state of the tumor, with less cycling cells, seemingly in contrast to the higher penetrance and decreased latency of tumor growth.

The authors then examined the rate of cell death in the tumor. Interestingly, Pten-null tumors had less dying cells, as assessed by TUNEL. In addition, the tumors expressed differentiaton markers NeuN and SyP, which are rare in SHH-MB mouse models. This reduction in dying cells is also evident at earlier stages of tumor growth. By looking shortly after Pten-loss induction, the authors found that Pten loss had an immediate impact on increasing the proliferative state of GCPs, followed by enhancing survival of differentiated cells. These two pro-tumor features together account for the increased penetrance and decreased latency of the model. While heterozygous loss of Pten also promoted proliferation, it did not protect against cell death.

Interestingly, loss of Pten alone in GCPs caused an increase in cerebellar size throughout development. The authors suggest that Pten normally constrants GCP proliferation, although they did not check whether reduced cell death is also contributing to cerebellum size.

Lastly, the authors examined macrophage infiltration and found that there was less macrophage infiltration to the Pten-null tumors. Using scRNA-seq, they suggest that the observed reduction in macrophages might be due to immunosuppressive tumor microenvironment.

This mouse model will be of high relevance to the medulloblastoma community, as current models do not reflect the heterogeneity of the disease. In addition, the elegant experimentation into Pten function may be relevant to cancer biologists outside of the medulloblastoma field.

Strengths:

The in-depth characterisation of the mouse model is a major strength of the study, including multiple time points and quantifications. The single-cell sequencing adds a nice molecular feature, and this dataset may be relevant to other researchers with specific questions of Pten function.

Weaknesses:

Adequately addressed in revisions.

---

## [Author Response]

The following is the authors’ response to the original reviews.

**eLife Assessment**
This valuable study provides insights into the role of Pten mutations in SHH-medulloblastoma, by using mouse models to resolve the effects of heterozygous vs homozygous mutations on proliferation and cell death throughout tumorigenesis. The experiments presented are convincing, with rigorous quantifications and orthogonal experimentation provided throughout, and the models employing sporadic oncogene induction, rather than EGL-wide genetic modifications, represent an advancement in experimental design. However, the study remains incomplete, such that the biological conclusions do not extend greatly from those in the extant literature; this could be addressed with additional experimentation focused on cell cycle kinetic changes at early stages, as well as greater characterization of macrophage phenotypes (e.g., microglia vs circulating monocytes). The work will be of interest to medical biologists studying general cancer mechanisms, as the function of Pten may be similar across tumor types.

We appreciate the summary of the importance of our work and agree that it provides a foundation for future experiments addressing underlying mechanisms including the role of macrophages in tumor progression/regression

**Public Reviews:**

**Reviewer #1 (Public review):**
Summary:This paper investigates how Pten loss influences the development of medulloblastoma using mouse models of Shh-driven MB. Previous studies have shown that Pten heterozygosity can accelerate tumorigenesis in models where the entire GNP compartment has MB-promoting mutations, raising questions about how Pten levels and context interact, especially when cancer-causing mutations are more sporadic. Here, the authors create an allelic series combining sporadic, cell-autonomous induction of SmoM2 with Pten loss in granule neuron progenitors. In their models, Pten heterozygosity does not significantly impact tumor development, whereas complete Pten loss accelerates tumour onset. Notably, Pten-deficient tumours accumulate differentiated cells, reduced cell death, and decreased macrophage infiltration. At early stages, before tumour establishment, they observe EGL hyperplasia and more pre-tumour cells in S phase, leading them to suggest that Pten loss initially drives proliferation but later shifts towards differentiation and accumulation of death-resistant, postmitotic cells. Overall, this is a well-executed and technically elegant study that confirms and extends earlier findings with more refined models. The phenotyping is strong, but the mechanistic insight is limited, especially with respect to dosage effects and macrophage biology.Strengths:The work is carefully executed, and the models-using sporadic oncogene induction rather than EGL-wide genetic manipulations-represent an advance in experimental design. The deeper phenotyping, including singlecell RNA-seq and target validation, adds rigor.Weaknesses:The biological conclusions largely confirm findings from previous studies (Castellino et al, 2010; Metcalf et al, 2013), showing that germline or conditional Pten heterozygosity accelerates tumorigenesis, generates tumors with a very similar phenotype, including abundant postmitotic cells, and reduced cell death.

We respectfully would like to point out that we have added new insights not covered in the previous more abbreviated studies. First, we are the first to show that in a sporadic model, heterozygous loss of Pten does not lead to accelerated or more aggressive disease. This is an important finding, since this is the case for many patients and only germline PTEN mutant humans are likely to have more aggressive tumors. Also, the previous studies did not examine tumor progress by analyzing neonatal stages or analyze spinal cord metastasis. We found a different phenotype at some early stages then at end stage, thus they provide new insights. Our study also is the only one to apply a mosaic analysis to study cell behaviors at early stages of progression, including proliferation and differentiation/survival. We are also the first to demonstrate a reduction in macrophages in Pten mutant SHH-MB.

The second stated goal - to understand why Pten dosage might matter - remains underdeveloped. The difference between earlier models using EGL-wide SmoA1 or Ptch loss versus sporadic cell-autonomous SmoM2 induction and Pten loss in this study could reflect model-specific effects or non-cell-autonomous contributions from Pten-deficient neighbouring cells in the EGL, for example. However, the study does not explore these possibilities. For instance, examining germline Pten loss in the sporadic SmoM2 context could have provided insight into whether dosage effects are cell-autonomous or dependent on the context.

We thank the reviewer for suggesting this experiment and agree it would be an informative one for other groups to perform as a follow up to our work to allow a direct comparison in the same sporadic SHH-MB model of mosaic vs germline loss of Pten. Also, we would like to point out that we do show a dosage effect of lowering vs removing Pten when only sporadic GCPs also have an activating mutation in SMO. Please see above comments for additional new mechanistic insight we have provided.

The observations on macrophages are intriguing but preliminary. The reduction in Iba1+ cells could reflect changes in microglia, barrier-associated macrophages, or infiltrating peripheral macrophages, but these populations are not distinguished. Moreover, the functional relevance of these immune changes for tumor initiation or progression remains unexplored.

We agree, further studies of the influence of Pten mutations on macrophage phenotypes will be interesting.

**Reviewer #2 (Public review):**
The authors sought to answer several questions about the role of the tumor suppressor PTEN in SHHmedulloblastoma formation. Namely, whether Pten loss increases metastasis, understanding why Pten loss accelerates tumor growth, and the effect of single-copy vs double-copy loss on tumorigenesis. Using an elegant mouse model, the authors found that Pten mutations do not increase metastasis in a SmoD2-driven SHH-medulloblastoma mouse model, based on extensive characterization of the presence of spinal cord metastases. Upon examining the cellular phenotype of Pten-null tumors in the cerebellum, the authors made the interesting and puzzling observation that Pten loss increased the differentiation state of the tumor, with fewer cycling cells, seemingly in contrast to the higher penetrance and decreased latency of tumor growth.The authors then examined the rate of cell death in the tumor. Interestingly, Pten-null tumors had fewer dying cells, as assessed by TUNEL. In addition, the tumors expressed differentiation markers NeuN and SyP, which are rare in SHH-MB mouse models. This reduction in dying cells is also evident at earlier stages of tumor growth. By looking shortly after Pten-loss induction, the authors found that Pten loss had an immediate impact on increasing the proliferative state of GCPs, followed by enhancing the survival of differentiated cells. These two pro-tumor features together account for the increased penetrance and decreased latency of the model. While heterozygous loss of Pten also promoted proliferation, it did not protect against cell death.Interestingly, loss of Pten alone in GCPs caused an increase in cerebellar size throughout development. The authors suggest that Pten normally constrains GCP proliferation, although they did not check whether reduced cell death is also contributing to cerebellum size.Lastly, the authors examined macrophage infiltration and found that there was less macrophage infiltration in the Pten-null tumors. Using scRNA-seq, they suggest that the observed reduction in macrophages might be due to an immunosuppressive tumor microenvironment.This mouse model will be of high relevance to the medulloblastoma community, as current models do not reflect the heterogeneity of the disease. In addition, the elegant experimentation into Pten function may be relevant to cancer biologists outside of the medulloblastoma field.Strengths:The in-depth characterisation of the mouse model is a major strength of the study, including multiple time points and quantifications. The single-cell sequencing adds a nice molecular feature, and this dataset may be relevant to other researchers with specific questions of Pten function.Weaknesses:One weakness of the study was the examination of the macrophage phenotype, which did not include quantification (only single images), so it is difficult to assess whether this reduction of macrophages holds true across multiple samples. Future studies will also be needed to assess whether Pten-mutated patient medulloblastomas also have a differentiation phenotype, but this is difficult to assess given the low number of samples worldwide.

We thank the reviewer for highlighting the importance of our sporadic mutant approach and new findings. As stated above, we agree, further studies of the influence of Pten mutations on macrophage phenotypes will be interesting as well as of human samples once large numbers can be obtained. All conclusions about macrophages are based on analyzing 3 independent tumors/genotype, which was stated in the Figure legends, and for all end stage tumors the sections were collected from one lateral edge of the tumor to the midline and for earlier stage from one side of the brain to the other, thus we believe the reported phenotypes are consistent within tumor and stages

**Recommendations for the authors:**

**Reviewer #1 (Recommendations for the authors):**
Minor points(1) The authors should state explicitly that early EGL analyses sample the same cerebellar region across animals (e.g., matched lobule or distance from the midline) because position-dependent effects are possible.

We agree this is an important aspect of the rigor of the study and are sorry this was not clear enough. We had stated in the legends to Figures 4 and 5 that midline sections were analyzed and when it was not the entire EGL quantified the region analyzed was shown, but we now include more details in all relevant Figure legends and in the Methods section.

(2) It is not clear from Figure 3i-k that TUNEL density in Syp-high regions differs between Pten+/- and Pten-/- tumors.

We have added a new graph as Figure 3 Supplemental Figure 1D with this direct comparison. Indeed, there is no difference between the Syp-high regions of Pten+/- and Pten-/- tumors as these regions of Pten+/- tumors have no detectable PTEN protein and thus have the same behavior as Pten-/- tumors (reduced cell death).

(3) The authors interpret the increase in the %EdU+ GFP+ cells in the EGL as evidence of a faster cell cycle. However, EdU labeling alone does not demonstrate altered cell cycle kinetics; this would require a dedicated assay. It would also be informative to combine EdU with Ki67 staining. This could clarify whether the effect reflects changes in differentiation - for example, if a higher proportion of GFP+ pre-tumor cells remain Ki67+-or whether the increase in EdU simply reflects a greater fraction of cells being in cycle. Such an analysis might even reveal no change in cycling if the proliferation index in controls is lower.

We are sorry we did not make our analysis sufficiently clear in Figure 5 and Figure 6. The quantification of EdU+ cells was restricted to the outer EGL (region defined by containing GFP+ and EdU+ cells) where all cells should be Ki67+. We cannot perform co-staining of Ki67 and GFP, since antigen retrieval for Ki67 removes the epitope for our GFP antibody. We have revised the wording in the figure legends and results sections.

(4) Some of the stains are unconvincing - for example, Figure 2 E,F, the p27 staining is difficult to distinguish from the background, Figure 7G,E- CD31+ blood vessels are difficult to see.

As requested, in Fig. 2 we adjusted the level of the green color for P27 to reduce the background in A, B, E , F using Photoshop. In Fig. 7G, H we adjusted the level of the green color for CD31 to reduce the background.

(5) Line 158: "unlike a SmoA2 model with germline or broad deletion of Pten in the cerebellum, where heterozygous deletion is sufficient..." That paper refers to the Neuro-D2SmoA1 mouse model. So this statement should be clarified.

We have made this edit.

**Reviewer #2 (Recommendations for the authors):**
(1) I find the final discussion paragraph about Kmt2d does not add much to the study, as it seems obvious that the mechanisms of tumor formation would differ between two different tumor suppressor genes, but this is only my opinion.

We respectfully think it is interesting, even if expected, so have left it in the Discussion.

(2) There is also a typo on line 342 that changes the meaning of the sentence: mTORC1 signaling is significantly 'unregulated';

We thank the reviewer for noticing this mistake. We have changed 'unregulated' to ‘upregulated’.

(3) Figure 9Q,R mislabeled: not mTORC1, but instead UPR

Asns is included in the mTOR pathway in Hallmark MTOR1 signaling as well as in the Unfolded Protein Response gene list. We have made a note of this in the Figure legend.